# ONE-SHOT REAL-WORLD DEMONSTRATION SYNTHESIS FOR SCALABLE BIMANUAL MANIPULATION

## ABSTRACT

Learning dexterous bimanual manipulation policies critically depends on large-scale, high-quality demonstrations, yet current paradigms face inherent trade-offs: teleoperation provides physically grounded data but is prohibitively labor-intensive, while simulation-based synthesis scales efficiently but suffers from sim-to-real gaps. We present BiDemoSyn, a framework that synthesizes contact-rich, physically feasible bimanual demonstrations from a single real-world example. The key idea is to decompose tasks into invariant coordination blocks and variable, object-dependent adjustments, then adapt them through vision-guided alignment and lightweight trajectory optimization. This enables the generation of thousands of diverse and feasible demonstrations within several hour, without repeated tele-operation or reliance on imperfect simulation. Across six dual-arm tasks, we show that policies trained on BiDemoSyn data generalize robustly to novel object poses and shapes, significantly outperforming recent baselines. By bridging the gap between efficiency and real-world fidelity, BiDemoSyn provides a scalable path toward practical imitation learning for complex bimanual manipulation without compromising physical grounding.

## 1 INTRODUCTION

Recent advances in robot manipulation have been propelled by data-driven imitation learning, where visuomotor policies trained on large-scale demonstration datasets enable dexterous single-arm and bimanual tasks previously deemed intractable. Methods like Action Chunking Transformer (ACT) Zhao et al. (2023a), Diffusion Policy (DP) Chi et al. (2023) and $\pi_0$ Black et al. (2024) exemplify this trend, achieving remarkable performance in contact-rich scenarios such as continuously pushing, battery assembly, and coordinated garment folding. These successes, however, hinge on a critical yet underappreciated premise: the availability of diverse, high-quality demonstrations, which are typically collected via expert teleoperation. As the community shifts toward more complex long-horizon bimanual tasks, the data scalability bottleneck becomes increasingly apparent. Even those largest manipulation datasets like RH20T Fang et al. (2024), DROID Khazatsky et al. (2024), Open X-Embodiment O'Neill et al. (2024) and AgiBot World Bu et al. (2025) struggle to provide sufficient geometric and kinematic diversity for robust generalization, underscoring a pivotal question: *how to scale real-world demonstration collection without compromising physical fidelity?*

The answer, unfortunately, lies in a fundamental trade-off. Existing data acquisition methods fall into two paradigms, each with critical shortcomings. On one hand, human-operated systems (*e.g.*, the ALOHA series Zhao et al. (2023a); Fu et al. (2024b); Zhao et al. (2024)) yield physically-accurate trajectories but demand prohibitive expertise, rendering them impractical for scaling to long-horizon tasks. On the other hand, simulation-based methods (*e.g.*, MimicGen Mandlekar et al. (2023), RoboGen Wang et al. (2024c), RoboCasa Nasiriany et al. (2024b), and DexMimicGen Jiang et al. (2024)) bypass human labor through programmatic data generation utilizing massive 3D assets, yet discrepancies in contact dynamics and visual rendering inevitably plague real-world deployment Bharadhwaj (2024); McCarthy et al. (2024). Thus, the tension between data quality (grounded in reality) and quantity (enabled by automation) persists as a key barrier to scalable imitation learning, particularly for contact-rich bimanual tasks requiring precise dual-arm coordination.

To bridge this gap, we present **BiDemoSyn** that synthesizes real-world bimanual demonstrations from a single exemplar, by algorithmically amplifying task semantics through a hierarchical decom-

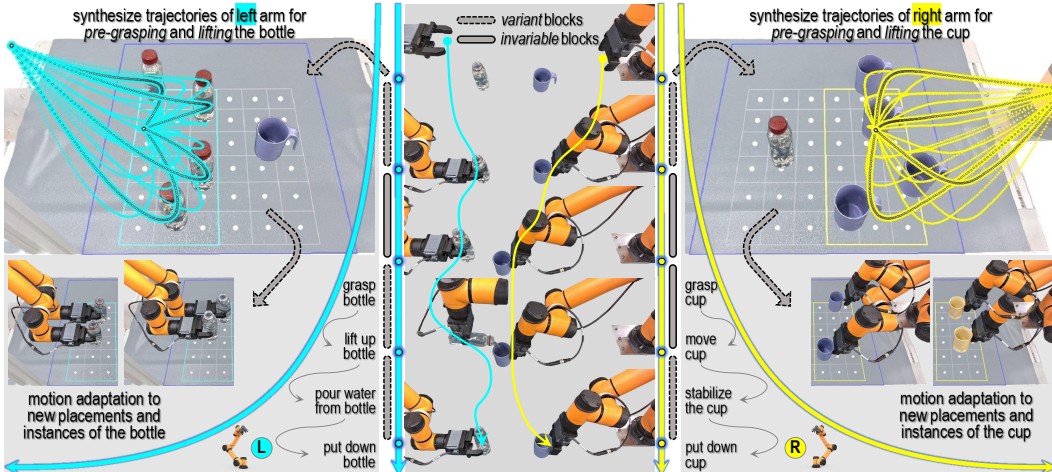

Figure 1: **From One to Many** ①→Ⓝ. Taking the example of dual-arm coordinated `pouring` task, we illustrate how to synthesize corresponding pre-grasping and lifting trajectories for different new placements and novel instances of manipulated objects during the initial frame alignment phase.

position of invariance and adaptability. Our key insight lies in recognizing that complex manipulation tasks exhibit a duality: while certain motion primitives (*e.g.*, stable grasp sequences, dual-arm synchronization) remain invariant across instances, others (*e.g.*, trajectory segments adapting to object shifting) require dynamic adjustments to contextual variations. Through a three-stage process, **BiDemoSyn** first deconstructs the initial demonstration into invariant primitives and geometry-sensitive components, then leverages visual perception to spatially align these elements with novel scenes, enabling generalization across object geometries and workspace layouts with minor human effort (primarily involving reinitializing manipulated objects). Finally, a physics-aware optimizer jointly modulates dual-arm trajectories, injecting diversity into adaptable components while enforcing real-world constraints on collision avoidance and kinematic coordination. This integrated approach achieves *one-to-many* demonstration synthesis: a single exemplar spawns hundreds of trajectories that preserve task intent while adapting to unseen configurations, all grounded in real-world dynamics rather than simulated proxies. Examples are shown in Fig. 1 and Fig. 2.

Unlike prior simulation-based methods Mandlekar et al. (2023); Wang et al. (2024c); Nasiriany et al. (2024b); Jiang et al. (2024), BiDemoSyn operates entirely within the physical domain, ensuring synthesized data inherits the fidelity of human demonstrations. By unifying algorithmic scalability with physical realism, our framework empowers policies to transcend the limitations of narrow manually-collected data distributions. Experiments on real dual-arm platforms validate the effectiveness of **BiDemoSyn**. And visuomotor policies trained with our synthesized demonstrations can achieve superior success rates and generalization in tasks requiring precise contact coordination (*e.g.*, small hole insertion, rotation angle control, purposeful handover and articulated object manipulation).

Our contributions are threefold: (1) **One-Shot Synthesis Framework**: A systematic pipeline combining task decomposition, vision-guided adaptation, and contact-aware trajectory optimization to generate scalable real-world bimanual demonstrations. (2) **Reality-Grounded Data Generation**: A completely simulator-free method for synthesizing bimanual demonstrations, ensuring physical fidelity by construction. (3) **Empirical Validation in Complex Tasks**: Comprehensive real-robot experiments demonstrating significant improvements in policy robustness and cross-configuration generalization on various bimanual manipulation tasks.

## 2 RELATED WORKS

**Bimanual Robotic Manipulation.** Prior literatures primarily target task-specific challenges (such as cloth folding Colomé & Torras (2018); Weng et al. (2022); Canberk et al. (2023), untangling Grannen et al. (2021); Peng et al. (2024), untwisting Lin et al. (2024a); Bahety et al. (2024), and handover Huang et al. (2023); Li et al. (2023)) through handcrafted controllers or narrow skill libraries, limiting generalization due to rigid motion primitives and excessive assumptions. General

methods Xie et al. (2020); Chitnis et al. (2020); Chen et al. (2022; 2023); Hartmann et al. (2022); Zhao et al. (2023b) either rely on predefined hierarchical arm roles (*e.g.*, leader-follower Krebs & Asfour (2022); Liu et al. (2022); Grotz et al. (2024) and stabilizer-actor Grannen et al. (2023); Liu et al. (2024a)) or hinge generalization entirely on dataset diversity, both of which are ultimately bottlenecked by the impractical costs of data acquisition. Recently, ALOHA series Zhao et al. (2023a); Fu et al. (2024b); Zhao et al. (2024) have revolutionized bimanual manipulation by dexterous low-cost teleoperation. Meanwhile, the visuomotor policy learning paradigm Chi et al. (2023); Ze et al. (2024); Yang et al. (2024a) based on diffusion models Ho et al. (2020); Song et al. (2021) has largely expanded the action modeling space and data utilization efficiency. Their followers Team et al. (2024); Kim et al. (2024); Liu et al. (2025); Black et al. (2024); Pertsch et al. (2025); Lin et al. (2025) expect to train universal policies using large-scale teleoperation data but face scalability barriers from human effort. To improve reachability and dexterity, some studies use specialized hardware (*e.g.*, multi-finger hands Wang et al. (2024a); Shaw et al. (2024); Fu et al. (2024a); Cheng et al. (2024b) or tactile sensors Lin et al. (2024b); Chen et al. (2024a)), which complicate real-world adoption. In contrast, we utilize a fixed-base dual-arm platform with parallel grippers, synthesizing diverse demonstrations from a single example. By optimizing coordination as dynamic constraints rather than predefined hierarchies, we bypass hardware specificity and data scalability limitations, enabling task-agnostic policies grounded in physical feasibility.

**Bimanual Demonstration Collection.** The dominant path for acquiring bimanual demonstrations is *human teleoperation* Khazatsky et al. (2024); Bu et al. (2025), which delivers high-quality data but suffers from prohibitive scalability costs. To alleviate it, two alternative routes have emerged: *simulation-based synthesis* Garrett et al. (2024); Hua et al. (2024); Liang et al. (2024); Yang et al. (2024c); Wang et al. (2024b) and *learning from human videos* Ponimatkin et al. (2025); Ye et al. (2025); Zhao et al. (2025); Kareer et al. (2024); Bharadhwaj et al. (2024). The former, exemplified by MimicGen Mandlekar et al. (2023) and DexMimicGen Jiang et al. (2024), augments a few demonstrations by generating variations in simulation using geometric transformations and kinematic constraints. Analogously, RoboGen Wang et al. (2024c) and RoboCasa Nasiriany et al. (2024b) leverage 3D assets to procedurally generate full-scene manipulation data, yet such methods inevitably inherit *sim-to-real* gaps, from unrealistic textures to unphysical dynamics. The latter extracts bimanual hand-object manipulation from egocentric videos Zhan et al. (2024); Liu et al. (2024b); Grauman et al. (2024) and maps them to dual arms via motion retargeting Li et al. (2024); Kerr et al. (2024); Chen et al. (2024b;c) or non-privileged representations (*e.g.*, keypoints Papagiannis et al. (2024); Gao et al. (2024); Wen et al. (2024b), affordances Ju et al. (2024); Nasiriany et al. (2024a) and correspondences Peng et al. (2024); Ko et al. (2024); Zhang & Boularias (2024)). While human videos are abundant and low-cost, large morphological mismatches between humanity and robotic embodiments often need heuristic translation rules and hinder direct applicability. Our work navigates these trade-offs by synthesizing demonstrations directly in real-world. Given a single exemplar, we diversify trajectories through vision-based adaptation and physics-compliant optimization. This balances the fidelity and scalability, while avoiding retargeting hurdles, thereby enabling practical and scalable data acquisition for contact-rich bimanual tasks.

Several concurrent works exhibit notable limitations. DemoGen Xue et al. (2025) and YOTO Zhou et al. (2025) diversify demonstrations via 3D editing of the initial single-view point cloud, but perspective ambiguities induce visual artifacts in synthesized data. ODIL Wang & Johns (2025) avoids editing by segmenting objects for visual servoing and iteratively replanning trajectories, which is cumbersome for scalability. MoMaGen Li et al. (2025) is a long-horizon mobile+bimanual demonstrations generator yet mainly focusing on the simulation domain. Our **BiDemoSyn** sidesteps these issues: synthesizing trajectories directly in real-world ensures visual-physical consistency without manual edits or robot replaying.

## 3 PRELIMINARIES

**Problem Formulation.** Bimanual imitation learning aims to train visuomotor policies $\pi_\theta(\mathbf{o}_t) \to \mathbf{a}_t$, where $\mathbf{o}_t$ denotes multimodal observations (*e.g.*, RGB images, joint states or point clouds) and $\mathbf{a}_t$ represents dual-arm actions. While architectures like ACT Zhao et al. (2023a) and DP Chi et al. (2023) enhance policy expressivity, their generalization critically depends on expert demonstrations $\mathcal{D} = \{\tau_i\}_{i=0}^N$ that densely span the feasible state-action manifold. Existing approaches either collect $\mathcal{D}$ via labor-intensive teleoperation or generate data in simulation, both failing to balance real-world fidelity and scalability. We address this gap by proposing the following problem:

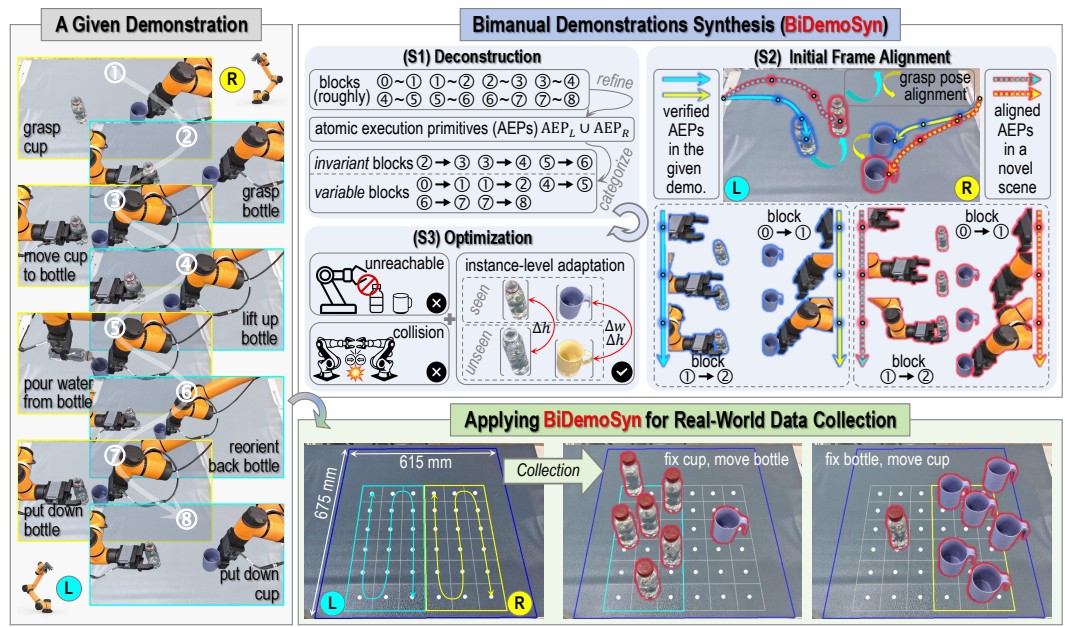

Figure 2: The overview of **BiDemoSyn**. It consists of three stages (*e.g.*, deconstruction, alignment, and optimization) based on a given demonstration. Then, we can apply our method to complete data collection efficiently and conveniently in real-world. It is best to zoom in to view the details.

Consider a bimanual task defined by an initial state $s_0$ and a specific goal $g$. Given a single real-world demonstration $\tau = \{\mathbf{o}_t, \mathbf{a}_t\}_{t=0}^T$, our objective is to synthesize a dataset $\mathcal{D}_{syn} = \{\tau_i\}_{i=0}^M (M \gg 1)$ such that: (1) **Task Consistency**: Every $\tau_i \in \mathcal{D}_{syn}$ achieves the task goal $g$ under perturbed initial states (*e.g.*, new object poses, scene layouts). (2) **Physical Admissibility**: Trajectories adhere to kinematic and dynamic constraints of the real-world system (*e.g.*, collision avoidance, dual-arm coordination). (3) **Diversity**: $\mathcal{D}_{syn}$ covers task-relevant variations to enable robust policy training.

**Visuomotor Policy Learning.** Our $\pi_\theta$ maps observations $\mathbf{o}_t$ to dual-arm actions $\mathbf{a}_t = [\mathbf{a}_t^L, \mathbf{a}_t^R] \in \mathbb{R}^{2d}$, where $\mathbf{a}_t^L, \mathbf{a}_t^R$ denotes left/right end-effector poses. We adapt a diffusion policy model Chi et al. (2023); Ze et al. (2024); Yang et al. (2024a) to generate action sequences $\mathbf{A} = [\mathbf{a}_{t:t+H}^L, \mathbf{a}_{t:t+H}^R] \in \mathbb{R}^{2H \times d}$ over a horizon $H$. The diffusion process iteratively denoises a noisy action sequence $\mathbf{A}_k$ toward kinematic feasibility over $K$ steps:

$$\mathbf{A}_k = \sqrt{\alpha_k}\mathbf{A}_0 + \sqrt{1-\alpha_k}\epsilon, \quad \epsilon \sim \mathcal{N}(0, \mathbf{\Sigma}), \tag{1}$$

where $\alpha_k$ follows a cosine schedule Nichol & Dhariwal (2021). And $\mathbf{\Sigma} \in \mathbb{R}^{2H \times 2H}$ is a block-diagonal covariance matrix $\begin{bmatrix} \mathbf{\Sigma}_L & \rho\mathbf{\Sigma}_{LR} \\ \rho\mathbf{\Sigma}_{LR}^\top & \mathbf{\Sigma}_R \end{bmatrix}$ encoding arm coordination priors, where $\mathbf{\Sigma}_L$, $\mathbf{\Sigma}_R$ governing temporal smoothness per arm, $\mathbf{\Sigma}_{LR}$ modeling inter-arm dependencies, and $\rho \in [0,1]$ controlling coordination strength. Then, a denoiser $\mu_\theta$ processes the noised sequence conditioned on visual observations $\mathbf{o}_t$ via:

$$\mu_\theta(\mathbf{A}_k, \mathbf{o}_t, k) = \mathtt{UNet}_\theta(\mathtt{Concat}\,[\mathbf{A}_k, \mathsf{E}(\mathbf{o}_t)]\,, k), \tag{2}$$

where $\mathsf{E}(\mathbf{o}_t)$ is a vision encoder (ResNet He et al. (2016) or PointNet Qi et al. (2017)) extracting latent features. The $\mathtt{UNet}_\theta$ employs asymmetric layers with early stages processing each arm independently, while deeper layers fusing bilateral coordinated features. Training minimizes:

$$\mathcal{L}(\theta) = \mathbb{E}_{\mathbf{A}_0, \mathbf{o}_t, k} \left[ \|\mu_\theta(\mathbf{A}_k, \mathbf{o}_t, k) - \mathbf{A}_0\|_{\mathbf{W}}^2 \right], \tag{3}$$

where $\|\cdot\|_{\mathbf{W}}$ is a Mahalanobis norm with $\mathbf{W} = \mathbf{\Sigma}^{-1}$ to prioritize arm interactions kinematically. By training on $\mathcal{D}_{syn}$ with feasible trajectories synthesized from a single exemplar, the policy bypasses sim-to-real gaps while handling diverse bimanual coordination patterns.

## 4 METHODOLOGY

Here, we detail **BiDemoSyn** for synthesizing bimanual demonstrations $\mathcal{D}_{syn}$ from a single exemplar $\tau$ (teleoperated or kinesthetically guided) with three stages: Sec. 4.1 **Deconstruction of One-Shot**

**Teaching**, which extracts invariant patterns and adaptable primitives; Sec. 4.2 **Vision-Based Initial Frame Alignment**, enabling generalization across geometric variations via efficient scene perception; and Sec. 4.3 **Trajectory Modulation and Optimization**, ensuring physical feasibility through hierarchical kinematic constraints. Each stage sequentially addresses scalability, adaptability, and real-world fidelity to bridge the gap between data efficiency and policy robustness. The overall framework is shown and illustrated in Fig. 2.

## 4.1 DECONSTRUCTION OF ONE-SHOT TEACHING

Given a single demonstration $\tau = \{\mathbf{o}_t, \mathbf{a}_t\}_{t=0}^T$, we decompose it into a *bimanual execution blocks* set $\{\mathcal{B}_i\}_{i=1}^n$, where each block $\mathcal{B}_i = (\mathbf{s}_i^L, \mathbf{s}_i^R)$ represents a discrete phase of dual-arm interaction. Here, $\mathbf{s}_i^L, \mathbf{s}_i^R \in \mathbb{R}^d$ denote the left/right arm state sequences (including end-effector 6-DoF poses and gripper status) within block $\mathcal{B}_i$. Blocks are categorized based on motion coordination. For the **single-arm motion**, one arm exhibits significant state transitions while the other remains static. Formally, for threshold $\delta$ (minimal motion saliency) and $\zeta$ (static tolerance):

$$(\left\|\mathbf{s}_{i,e}^L - \mathbf{s}_{i,s}^L\right\| \geq \delta \wedge \left\|\mathbf{s}_{i,e}^R - \mathbf{s}_{i,s}^R\right\| \leq \zeta) \ \vee \ (\left\|\mathbf{s}_{i,e}^R - \mathbf{s}_{i,s}^R\right\| \geq \delta \wedge \left\|\mathbf{s}_{i,e}^L - \mathbf{s}_{i,s}^L\right\| \leq \zeta). \tag{4}$$

While, for the **dual-arm coordination**, both arms synchronously or asynchronously adjust states:

$$\left\|\mathbf{s}_{i,e}^L - \mathbf{s}_{i,s}^L\right\| \geq \delta \wedge \left\|\mathbf{s}_{i,e}^R - \mathbf{s}_{i,s}^R\right\| \geq \delta. \tag{5}$$

**From Blocks to Atomic Execution Primitives (AEPs)**. To enable modular adaptation, each block $\mathcal{B}_i$ is further refined into a minimal motion unit AEP, where an arm undergoes a salient state transition (*e.g.*, gripper closure, moving or pose shifts). For arm $A \in \{L, R\}$, an AEP is defined as:

$$\mathrm{AEP}_A = \left\{\mathbf{s}_t^A \rightarrow \mathbf{s}_{t+\Delta t}^A \mid \left\|\mathbf{s}_{t+\Delta t}^A - \mathbf{s}_t^A\right\|_2 \geq \gamma, \Delta t \leq T_{\max}, \mathtt{CollisionFree}(\mathbf{s}_{[t,\Delta t]}^A)\right\}, \tag{6}$$

where $\gamma$ is a distance threshold, $T_{\max}$ limits execution duration, and $\mathtt{CollisionFree}(\cdot)$ enforces no collisions between the arm and objects/scene (optimized via subsequent forward kinematics). Unlike keyframe-based segmentation James et al. (2022); Shridhar et al. (2023); Ma et al. (2024), AEPs ignore acceleration profiles (assuming quasi-static motions) and prioritize task-oriented transitions over temporal granularity.

**Semantic Categorization for Adaptive Synthesis**. The final step categorizes refined blocks into *invariant* and *variable* types. The invariant blocks encode task-semantic primitives (*e.g.*, screwing, pressing) or general motions (*e.g.*, lifting, transferring), which remain structurally consistent across variations. The variable blocks, such as object-centric goal-conditioned grasping, adapt to instance-level geometric variations including object poses and shapes through:

$$\mathcal{B}_i' = \mathcal{B}_i \circ \Phi(g, \mathbf{o}_{\mathrm{novel}}), \tag{7}$$

where $\Phi(\cdot)$ is a vision-driven adapter, $g$ is the task goal, and $\mathbf{o}_{\mathrm{novel}}$ is the novel observation. This structured decomposition enables selective diversification, allowing variable blocks to adapt to new scenarios while preserving task fidelity in invariant blocks.

## 4.2 VISION-BASED INITIAL FRAME ALIGNMENT

The variable blocks identified in Sec. 4.1 (*e.g.*, initial grasping) require adapting to novel object poses and geometries. To achieve this, we propose a vision-driven adapter $\Phi(\cdot)$ that generalizes object-centric interactions in the given exemplar to new scenes. Given a novel observation $\mathbf{o}_{\mathrm{novel}}$, our method executes three steps: **object perception**, **state estimation**, and **pose alignment**, ensuring the initial robotic grasp aligns with the task goal $g$. Illustrations are shown in Fig. 3.

**Object Perception**. We first detect and segment the target object in $\mathbf{o}_{\mathrm{novel}}$ using an open-vocabulary detector (*e.g.*, YOLO-World Cheng et al. (2024a) or YOLOE Wang et al. (2025)) or vision foundation models (*e.g.*, Florence2 Xiao et al. (2024) with SAM2 Ravi et al. (2024) for robustness to rare categories). In practice, we prioritize foundation models to avoid detection failures. Now let $M \subseteq \mathbf{o}_{\mathrm{novel}}$ denote the obtained object's binary mask.

**State Estimation**. Instead of uitlizing category-level CAD-based 6D pose estimation models (*e.g.*, FoundationPose Wen et al. (2024a)), we estimate the instance-level 6D object pose $\mathbf{P}_{\mathrm{obj}} \in SE(3)$

Figure 3: Illustrations of the initial frame alignment stage applied to tasks `pouring` (left and middle) and `reorient` (right). It shows that we can automatically adjust the grasp pose after the position, orientation and shape of the manipulated object changes.

using geometry-aware processing. Sepcifically, we adopt classic image moments Chaumette (2004); Kotoulas & Andreadis (2007) to calculate the object mask centroid $\mathbf{c}$ and extract principal axes $\mathbf{R}$:

$$
\begin{cases}
\mathbf{c} &= \frac{1}{|M|} \sum_{(u,v) \in M} (u, v, \mathrm{d}(u,v)) \\
\mathbf{R} &= \texttt{PCA}(\{(u, v, \mathrm{d}(u,v)) \mid (u,v) \in M\})
\end{cases}
\tag{8}
$$

where $\mathrm{d}(u,v)$ is the depth value. `PCA` is used to fit 3D points of $M$ to determine orientation $\mathbf{R} \in SO(3)$. This yields $\mathbf{P}_{\text{obj}} = (\mathbf{R}, \mathbf{c})$, robust to arbitrary object states (e.g., fallen, inverted).

**Pose Alignment**. Let $\mathbf{P}_{\text{demo}}$ denote the object pose in the given demonstration. We compute the rigid transformation $\mathbf{T} \in SE(3)$ that maps $\mathbf{P}_{\text{demo}}$ to $\mathbf{P}_{\text{obj}}$. This transformation is applied to the initial grasp pose $\mathbf{P}_{\text{grasp,demo}}$ in the variable block $\mathcal{B}_i$, yielding the adapted grasp pose:

$$
\mathbf{P}_{\text{grasp,novel}} = \mathbf{T} \cdot \mathbf{P}_{\text{grasp,demo}} = (\mathbf{P}_{\text{obj}} \cdot \mathbf{P}_{\text{demo}}^{-1}) \cdot \mathbf{P}_{\text{grasp,demo}}.
\tag{9}
$$

The robot arm then executes a collision-free motion to reach $\mathbf{P}_{\text{grasp,novel}}$, ensuring goal-conditioned grasping without relying on task-irrelevant dense grasp proposals produced via 6-DoF grasp pose detectors Fang et al. (2020; 2023). By integrating with the decomposition stage, the adapted grasp pose directly modifies the variable block $\mathcal{B}_i$ into $\mathcal{B}_i'$, enabling synthesis of new trajectories.

## 4.3 TRAJECTORY MODULATION AND OPTIMIZATION

After deconstructing the demonstration into blocks (Sec. 4.1) and adapting variable blocks via vision-based alignment (Sec. 4.2), the synthesized trajectory requires two critical refinements to ensure executability and robustness as below:

**Collision-Aware Validation**. Each adapted block undergoes kinematic feasibility checks. First, we validate the reachability of target poses $\mathbf{P}_{\text{grasp,novel}}$ using Inverse Kinematics (IK), ensuring the robot joint limits are satisfied. If the result returned is singular or unreachable (rarely happens), we will delete this sample. Second, a motion planner Chitta et al. (2012); Schulman et al. (2014) verifies potential collision between arms and scene objects by solving for collision-free paths between the start and end states in each block. This simplified two-point constraint reduces computational overhead while preserving safety, assuming the trajectory of original given demonstration is collision-free.

**Instance-Level Motion Adaptation**. To handle geometric variations across object instances with differing shapes and sizes, we adjust motion primitives in variable blocks based on the object 3D bounding box dimensions. Let $\Delta l$, $\Delta w$ and $\Delta h$ denote the length, width and height differences between the novel object bounding box $\mathbf{b}_{\text{novel}}$ and the original $\mathbf{b}_{\text{demo}}$. Motion endpoints (*e.g.*, grasp or release positions) are offset by:

$$
\mathbf{s}_{i,e}^{A'} = \mathbf{s}_{i,e}^{A} + \lambda(\Delta l, \Delta w, \Delta h),
\tag{10}
$$

where $\lambda$ scales the adjustment (empirically set to $0.8 \sim 1.0$), and $A \in \{L, R\}$. For irregular shapes, depth-aware masks $M$ extracted in Sec. 4.2 can approximate volumetric differences, or minimal human input provides precise measurements.

This hierarchical optimization stage operates at the block level rather than full trajectories, and enables efficient scaling while preserving the task semantics encoded in invariant blocks. Finally, validated and adapted blocks are recombined into synthetic trajectories $\tau' = \{\tau_{\text{inv}}, \{\mathcal{B}_i'\}\}$, populating diverse and physically feasible demonstrations $\mathcal{D}_{syn}$.

## 5 EXPERIMENTS

Our experiments address three core questions. Q1: Is BiDemoSyn truly efficient and user-friendly? Q2: Does BiDemoSyn enable scalable visuomotor imitation learning? Q3: Does synthetic demonstrations generalize to spatial and object variations?

### 5.1 EXPERIMENTAL SETUP AND PROTOCOL

**Tasks**. We evaluate on six bimanual manipulation tasks (refer Fig. 9) requiring contact-rich coordination: `plugpen`, `inserting`, `unscrew`, `pouring`, `pressing` and `reorient`. These tasks cover diverse primitive skills (*e.g.*, grasping, rotating and handover), and involve both rigid and articulated objects. The one-shot demonstration for each task is collected via kinesthetic teaching. For the platform (refer Fig. 8), two fixed-base arms equipped with parallel grippers are used. Scene perception is provided by a stereo camera capturing binocular images. More details of all tasks and the corresponding data collection process are in Appendix Sec. A.

**Policies & Baselines**. For a comprehensive comparison, baselines contain two categories. One category is *purely for data collection* including point cloud editing (DemoGen Xue et al. (2025)), real robot auto-rollout (YOTO Zhou et al. (2025)), and human drag teaching (close to Teleoperation). Generally, the quality of collected demonstrations by these baselines is better in turn, but the cost is more time-consuming. After preparing the training data, we adapt three advanced visuomotor policies (DP Chi et al. (2023), DP3 Ze et al. (2024) and EquiBot Yang et al. (2024a)) to bimanual settings by modifying their modeling spaces to dual-arm actions (refer Sec. 3). Observation inputs are RGB-only images or segmented 3D point clouds of task-relevant objects. Final trained policies operate in the open-loop discrete keyposes prediction to align with our synthesized demonstration format. The another category is *directly for bimanual manipulation without retraining* including the zero-shot ReKep Huang et al. (2024), an advanced ReKep+ with oracle-level grasp labels at the beginning, and one-shot ODIL Wang & Johns (2025).

**Metrics**. We evaluate (1) *synthesis efficiency*: average time per demo over 50 tests, (2) *data quality*: visual authenticity of newly acquired observations, (3) *success rates*: real robot deployment with 30 trials per task, and (4) *generalization*: performance on unseen object placements and geometries.

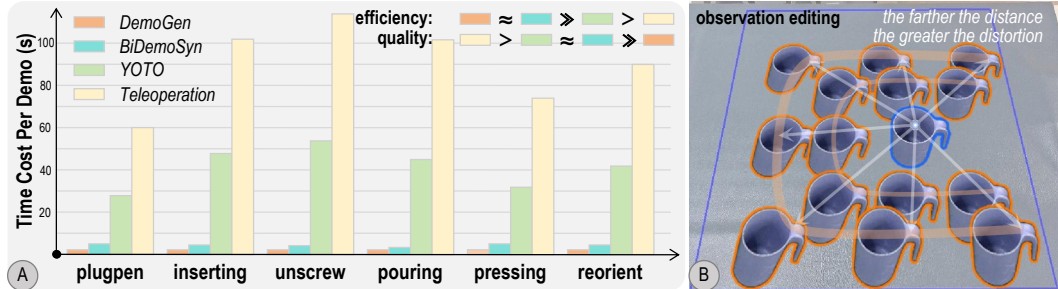

Figure 4: (A) The data collection efficiency comparison of different baselines and our BiDemoSyn, and (B) the generated data quality illustration of DemoGen. Although DemoGen has the highest synthesis efficiency, it cannot avoid visual artifacts caused by perspective transformation, so its data quality is the lowest. Our method can achieve a balance between the speed and quality.

### 5.2 COMPARISON AND PRESENTATION OF RESULTS

We here answer three questions raised earlier, demonstrating that BiDemoSyn can efficiently synthesize high-quality real-world demonstrations, enabling scalable and generalizable visuomotor policy training with minimal human input. Then there are some snapshots of real robot execution effects.

(A1): **The efficiency and usability of BiDemoSyn have obvious advantages over baselines**. Without bells and whistles, our BiDemoSyn demonstrates superior efficiency and usability as illustrated in Fig. 4. It requires *about 5 seconds per demonstration* to synthesize new trajectories, where primarily involving manual object repositioning and visual verification of initial frame alignment, followed by the trajectory optimization. In contrast, DemoGen generates trajectories in *less then 1 second* via scripted edits but suffers from perspective distortion artifacts, degrading data quality. The specific qualitative effect can be compared with Fig. 4A and our results in Fig. 2 and Fig. 9. By relying on manual alignment and auto-replay, YOTO takes *42 seconds per demo* which is equivalent to

Table 1: Quantitative comparison results of success rates for different methods under *in-distribution* and *out-of-distribution* evaluations. The # means the BiDemoSyn training data size.

| Policy (Input Mode) | Method | *in-distribution* evaluations | | | | | | | *out-of-distribution* evaluations | | | | | | |
| --- | --- | --- | --- | --- | --- | --- | --- | --- | --- | --- | --- | --- | --- | --- | --- |
| | | plugpen #3888 | inserting #7776 | unscrew #1152 | pouring #2592 | pressing #1296 | reorient #1008 | Average Success Rate | plugpen #2916 | inserting #3888 | unscrew #1008 | pouring #972 | pressing #324 | reorient #756 | Average Success Rate |
| Training-Free (RGB) | ReKep | 15/30 | 13/30 | 12/30 | 15/30 | 12/30 | 09/30 | 42.2% | 12/30 | 10/30 | 10/30 | 09/30 | 09/30 | 06/30 | 31.1% |
| | ReKep+ | 17/30 | 17/30 | 14/30 | 19/30 | 20/30 | 11/30 | 54.4% | 15/30 | 14/30 | 12/30 | 12/30 | 11/30 | 09/30 | 40.6% |
| | ODIL | 18/30 | 19/30 | 13/30 | 18/30 | 18/30 | 12/30 | 54.4% | 12/30 | 14/30 | 08/30 | 11/30 | 13/30 | 08/30 | 36.7% |
| DP (RGB) | DemoGen | 18/30 | 20/30 | 16/30 | 16/30 | 14/30 | 15/30 | 55.0% | 08/30 | 03/30 | 10/30 | 04/30 | 03/30 | 07/30 | 19.4% |
| | YOTO | 19/30 | 20/30 | 16/30 | 17/30 | 14/30 | 16/30 | 56.7% | 08/30 | 04/30 | 10/30 | 05/30 | 04/30 | 07/30 | 21.1% |
| | **BiDemoSyn** | 22/30 | 24/30 | 22/30 | 21/30 | 17/30 | 19/30 | **67.8%** | 13/30 | 10/30 | 18/30 | 15/30 | 08/30 | 12/30 | **42.2%** |
| DP3 (PCD) | DemoGen | 21/30 | 24/30 | 19/30 | 20/30 | 18/30 | 17/30 | 66.1% | 11/30 | 06/30 | 14/30 | 07/30 | 05/30 | 11/30 | 30.0% |
| | YOTO | 22/30 | 23/30 | 20/30 | 20/30 | 18/30 | 19/30 | 67.8% | 12/30 | 06/30 | 15/30 | 08/30 | 06/30 | 14/30 | 33.9% |
| | **BiDemoSyn** | 26/30 | 28/30 | 25/30 | 24/30 | 21/30 | 22/30 | **81.1%** | 16/30 | 14/30 | 21/30 | 21/30 | 11/30 | 18/30 | **54.4%** |
| EquiBot (PCD) | DemoGen | 22/30 | 24/30 | 20/30 | 20/30 | 19/30 | 19/30 | 68.9% | 13/30 | 10/30 | 16/30 | 11/30 | 10/30 | 13/30 | 40.6% |
| | YOTO | 23/30 | 24/30 | 20/30 | 21/30 | 20/30 | 19/30 | 71.1% | 15/30 | 11/30 | 18/30 | 14/30 | 12/30 | 16/30 | 47.8% |
| | **BiDemoSyn** | 28/30 | 28/30 | 26/30 | 25/30 | 24/30 | 25/30 | **86.7%** | 18/30 | 20/30 | 24/30 | 21/30 | 17/30 | 20/30 | **66.7%** |

real robot execution time. While, teleoperation demands *91 seconds per demo* for near-perfect but labor-intensive collection. These results highlight BiDemoSyn's unique balance of speed (automated optimization) and reliability (artifact-free synthesis), making it the only method scalable to thousands of demonstrations without compromising real-world fidelity.

(A2): **Demonstrations obtained via BiDemoSyn can support scalable imitation learning**. We applied BiDemoSyn for efficient collection of thousands demonstrations per task (Fig. 9 right), densely covering the workspace with instance-level object diversity, thereby providing abundant training data for imitation learning. For two reproduced baselines, DemoGen Xue et al. (2025) matches our data scale but suffers from visual artifacts, while YOTO Zhou et al. (2025) collects only 1/10 the data per task for being limited by its time-consuming replay mechanism. After obtaining adequate data, we trained three advanced visuomotor policies DP Chi et al. (2023), DP3 Ze et al. (2024) and EquiBot Yang et al. (2024a). As shown in Tab. 1 left, BiDemoSyn achieves superior *in-distribution* (ID) success rates across all six tasks,

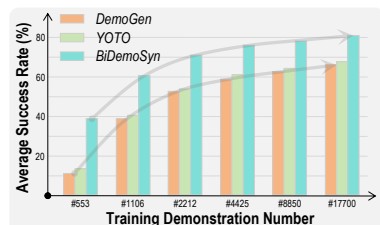

Figure 5: Comparison between training data scale and the success rate. Less sized data is randomly sampled out of the total dataset at the task level. DP3 is chosen as the visuomotor policy.

outperforming both baselines regardless of the policy architecture. This confirms that BiDemoSyn's artifact-free synthesis and efficient data scaling reliably support the scaling laws of imitation learning. Besides, it is unsurprising that policies trained with demonstrations consistently outperform the two training-free baselines ReKepHuang et al. (2024) and ODIL Wang & Johns (2025). To further quantify scalability, we evaluated training data size-performance relationships on all six tasks of all methods. As shown in Fig. 5, BiDemoSyn-trained policies exhibit stronger scaling trends than DemoGen (e.g., 60% vs. 39% at 1106 demos), aligning with community observations that data quality and quantity jointly drive visuomotor policy performance Liu et al. (2025); Black et al. (2024); Pertsch et al. (2025); Lin et al. (2025).

(A3): **Policies trained on BiDemoSyn data can achieve better generalization to unseen variations**. To evaluate the generalization to novel instances, we conduct controlled tests across all six tasks. For each task, we exclude demonstrations associated with one or a pair of randomly selected objects (*e.g.*, an arbitrary bottle in unscrew or a bottle-cup pair in pouring) from the training set, resulting in training

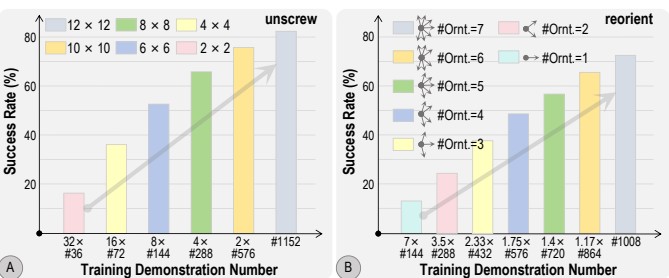

Figure 6: Analysis of spatial generalization, including variations in (A) position sampling density and (B) orientation sampling diversity. DP3 is chosen as the visuomotor policy.

sizes of 2916/3888/1008/972/324/756 demos for six tasks, respectively. These excluded objects are

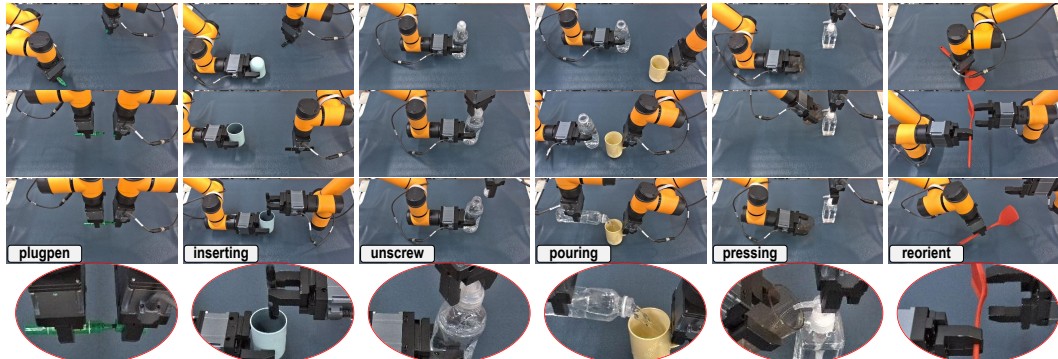

Figure 7: Visualization of all six bimanual tasks performed on real robots. All models are trained and tested under the ID evaluations. EquiBot is chosen as the visuomotor policy. Key dual-arm coordination movements associated with each task are partially enlarged for quick review.

reintroduced exclusively during real-world testing, with identical data splits applied to DemoGen and YOTO baselines for fair comparison. As shown in Tab. 1 right, policies trained on BiDemoSyn data consistently achieve higher success rates compared to all baselines in *out-of-distribution* (OOD) settings. While all methods exhibit performance drops relative to ID evaluations, BiDemoSyn's superior OOD robustness highlights its ability to capture task-invariant features through vision-aligned synthesis. To further analyze spatial generalization, we vary the density of positional and orientation coverage in training data: for `unscrew`, we synthesize trajectories with sparse-to-dense workspace coverage (Fig. 6A), and for `reorient`, we incrementally increase the diversity of object orientations (Fig. 6B). To exclude the effect of data size differences, the total number of train-set is always augmented to a standard size (*e.g.*, `#1152` and `#1008`) via the DemoGen to edit some observations at small distances. Finally, policies trained on BiDemoSyn data with dense spatial coverage (*e.g.*, $10 \times 10$ workspace coverage in `unscrew`) achieve 76.7% success on unseen positions, compared to 53.3% for sparse coverage ($6 \times 6$ grid cells). Similarly, orientation diversity in `reorient` improves generalization from 37.3% (lower diversity with `#Ornt.=3`) to 73.3% (higher diversity with `#Ornt.=7`). These results confirm that BiDemoSyn's ability to cheaply synthesize positionally and orientationally diverse demos directly translates to enhanced policy generalization, aligning with empirical insights that broad data coverage mitigates spatial distribution shifts.

**Qualitative Results**: Real rollouts in Fig. 7 showcase effectiveness across six tasks. For example, the `plugpen` policy precisely aligns and inserts the pen into the narrow cap even with ±5mm positional noise. In `inserting`, dual-arm coordination achieves rough but proximate peg-in-hole precision. The `unscrew` demonstrates robust rotational synchronization, successfully loosening lids with variable thread tightness. The `pouring` policy adapts liquid flow control to container geometries, minimizing spills despite tilted orientations. For `pressing`, predicted actions reliably activate nozzles without over-pressuring, while `reorient` handles object handovers and flips with smooth dual-arm transitions. These visualizations corroborate our quantitative findings, emphasizing BiDemoSyn's ability to handle real-world complexity in contact-rich manipulation.

## 6 CONCLUSION AND LIMITATION

We present BiDemoSyn, a framework for synthesizing real-world bimanual demonstrations from a single human example. By decomposing tasks into invariant and adaptable components, leveraging vision-guided scene alignment, and hierarchically optimizing trajectories, BiDemoSyn generates diverse, collision-free data without synthetic gaps. Experiments on six complex contact-rich tasks demonstrate its efficiency, scalability, and generalization, outperforming strong baselines and pure teleoperation. This work establishes a new paradigm for scalable imitation learning, bridging the divide between data quantity and physical fidelity in bimanual robotic manipulation.

**Limitations**: While BiDemoSyn excels in static, geometrically varied scenes, it faces challenges in dynamic environments and extreme shape variations (*e.g.*, deformable fabric objects). The current hierarchical optimization assumes quasi-static motions, limiting its applicability to highly dynamic tasks like catching. Additionally, synthesizing trajectories for multi-stage tasks with interdependent contacts (*e.g.*, lacing ropes) requires further extension. Future work will integrate dynamic perception and adaptive contact modeling to address these constraints.

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

APPENDIX

This supplementary part provides detailed explanations and extensions to the main paper. Sec. A outlines the design rationale for our six bimanual manipulation tasks, including platform specifications, task-specific objectives, and one-shot demonstration acquisition protocols. Sec. B details the data collection standards, UI-assisted collection tools, and post-processing workflows. These constitute the trajectory synthesis pipelines for generating real-world demonstrations with BiDemoSyn. We have also added more details about BiDemoSyn to help readers better understand this study. Sec. C elaborates on policy training and deployment, covering implementation specifics for baseline methods (DemoGen Xue et al. (2025), YOTO Zhou et al. (2025)) and diffusion policies (DP3 Ze et al. (2024), EquiBot Yang et al. (2024a)), alongside real-robot deployment. Sec. D provides additional visualizations of real-robot executions and failure case analyses, offering insights into current limitations. Sec. E concludes with reflections and future directions, aiming to extend BiDemoSyn to challenge more complex tasks (*e.g.*, deformable or dynamic manipulation) and enhance cross-domain generalization. Sec. F is the statement of the use of Large Language Models.

## A  DESIGN OF BIMANUAL MANIPULATION TASKS

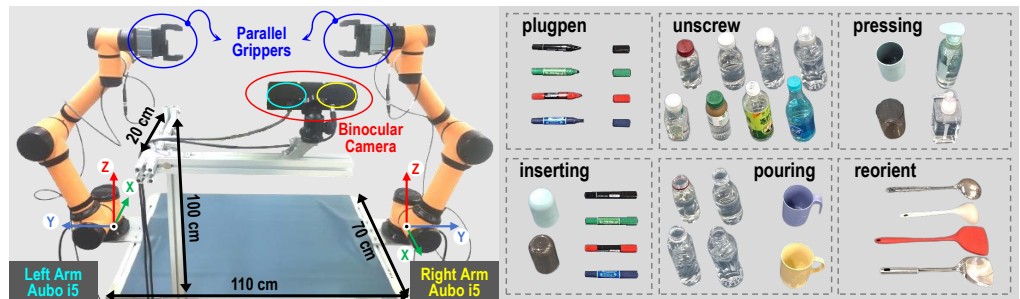

Figure 8: *Left*: The fixed-base dual-arm manipulator platform (a table with two robot arms, two grippers and the binocular camera) used in this research. *Right*: The object assets involved in our six bimanual manipulation tasks. All objects have been scaled down proportionally.

### A.1  HARDWARE AND PLATFORM

Our experimental platform comprises a rectangular workspace (110cm×70cm) with two 6-DoF AUBO-i5 collaborative arms[1] (880mm reach) mounted on opposite short edges of the table (refer Fig. 8 left). This opposing-arm configuration maximizes shared workspace while minimizing self-collision risks, albeit differing from anthropomorphic designs. Each arm is equipped with a DH-Robotics parallel gripper[2] (80mm max opening, 50mm effective length), controlled in binary states (open/closed). Tool length compensation accounts for 160mm absolute length of the gripper. The scene perception is provided by a binocular stereo Kingfisher R-6000 camera (960×540 RGB resolution), mounted 100cm above the table long edge to capture a third-person view of the workspace. The calibrated stereo setup reconstructs high-fidelity 3D point clouds Xu et al. (2023), eliminating the need for wrist-mounted cameras while ensuring full task visibility.

### A.2  BIMANUAL TASK FORMULATION

We design six bimanual manipulation tasks (including `plugpen`, `inserting`, `unscrew`, `pouring`, `pressing` and `reorient`) to comprehensively evaluate dual-arm coordination and generalization. Each task involves at least two category-level object instances (refer Fig. 8 right) and requires diverse primitive skills, such as single-arm actions (grasping, placing, carrying, precision rotation) and dual-arm coordination (fixed-twisting, plug-in, handover, alignment). All tasks are **long-horizon**, starting from goal-conditioned initial grasping (objects fully separated from robots), contrasting prior works already grasping/holding the manipulated object and focusing on short-horizon atomic skills such as untwisting Lin et al. (2024a); Bahety et al. (2024) or handover Huang et al. (2023). Fig. 9 shows some examples. Below we detail each task:

---

[1]https://www.aubo-cobot.com/public/i5product3

[2]https://en.dh-robotics.com/product/pg

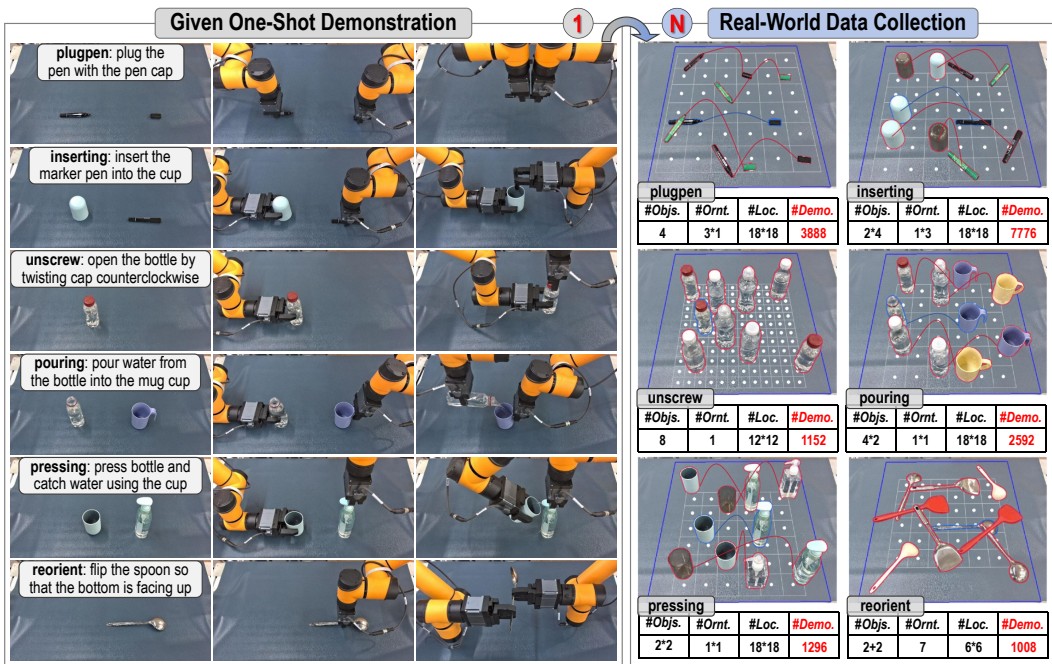

Figure 9: **From One to Many** ①→Ⓝ. *Left*: Six representative bimanual manipulation tasks with their one-shot demonstrations and task-specific descriptors. *Right*: Real-world data collection diagrams, showing object instances with varied geometries and spatial arrangements used to synthesize diverse demonstrations (*e.g.*, thousands physically consistent trajectories per task).

- plugpen: *plug the pen with the pen cap*. **States**: A marker pen body and its cap are placed separately on the table. The pen body lies horizontally with its tip roughly pointing to the right arm, while the cap is also positioned horizontally with its socket facing the left arm. **Steps**: Left arm grasps pen body; right arm grasps cap. Arms lift and spatially align the pen tip with the cap's socket before insertion. **Challenges**: Directional grasping alignment, sub-millimeter positional tolerance for insertion.

- inserting: *insert the marker pen into the cup*. **States**: An empty handle-free cup is placed upside-down on the table, and a closed marker pen lies horizontally nearby. **Steps**: Left arm flips the cup (about 180° rotation); right arm reorients the marker vertically. Arms coordinate to align and insert the marker into the upright cup. **Challenges**: Dual-arm rotational synchronization, tight insertion tolerance ($\pm$2mm).

- unscrew: *open the bottle by twisting cap counterclockwise*. **States**: A translucent plastic bottle (filled with colorless water) stands upright on the table, with its cap tightly screwed on. **Steps**: Left arm stabilizes the bottle; right arm grasps the cap and rotates it vertically (with multiple degree-fixed rotations). **Challenges**: Force-agnostic cap grasping (no torque sensing), rotational precision ($\pm$5° per turn).

- pouring: *pour water from the bottle into the mug cup*. **States**: An uncapped water bottle (about 3/4 full) and an empty handled mug are placed on opposite sides of the table. **Steps**: Left arm tilts the bottle (about 90°); right arm positions the mug to catch water. **Challenges**: Fluid dynamics approximation, nozzle-cup alignment under gripper deflection.

- pressing: *press bottle and catch water using the cup*. **States**: A shampoo bottle (with a pressable nozzle) containing water and an upright empty handle-free cup are placed apart. **Steps**: Right arm vertically presses the nozzle; left arm tilts the cup to catch water. **Challenges**: Nozzle positioning accuracy ($\pm$5mm), open-loop force control.

- reorient: *flip the spoon so that the bottom is facing up*. **States**: A long spoon or shovel (metal or plastic) lies concave-down on the table, with randomized orientation and position. **Steps**: Right arm grasps the spoon center, reorients it mid-air, and hands it to the left arm, which then flips and places it. **Challenges**: Unstable grasp during handover, rotational precision for flipping ($\pm$10°).

These tasks collectively stress *spatial reasoning*, *contact-aware coordination*, and *generalization* to *instance-level variations*, which are cornerstones of real-world bimanual manipulation. Besdies,

it is important to note that, we have **shared hardware constraints**: (1) No force/torque sensing limits contact-rich adjustments (*e.g.*, pen plug-in force, cap twisting force, or pressing force). (2) Binary gripper states (open/closed) restrict fine-grained manipulation (*e.g.*, plastic bottles are always deformed by excessive clamping). (3) Third-person view occlusions occasionally hinder precise alignment (*e.g.*, cannot discern the groove side of the pen cap).

### A.3 Obtaining and Decomposing the One-Shot Demonstration

As stated in the main content, we collect one-shot demonstrations via kinesthetic teaching: an operator manually guides both arms through task-critical waypoints, recording the 6DoF end-effector poses (relative to each robot base frame) and gripper binary states at each pause. Objects are placed in fixed initial configurations (allowing small positional tolerance) to ensure consistency. The recorded waypoints are then executed autonomously by the robot control API, which solves inverse kinematics between consecutive given poses and synchronizes gripper actions (*e.g.*, closing after reaching a pre-grasp pose). During auto-execution, stereo camera observations (10Hz) and dual-arm joint/end-effector states are logged. For the real rollout effect of the one-shot demonstration related to each task, please refer to our **Supplementary Videos**. Finally, demonstrations are deconstructed into task-aware blocks to support subsequent trajectory synthesis.

To enable reliable downstream synthesis, the one-shot kinesthetic demonstration must be collected in a way that naturally exposes task-relevant structure. During kinesthetic teaching, the demonstrator physically guides the two manipulators through the task while pausing at intuitive semantic boundaries, such as after a stable grasp, after completing an alignment adjustment, or before switching from transportation to interaction. These pauses, which arise organically from human motion patterns, produce clear temporal discontinuities that align well with the block-based representation introduced in Sec. 4.1 of the main text. Importantly, this process *does not require any manual annotation or labeling*. The demonstration is captured in a continuous stream exactly as in standard kinesthetic teleoperation, and the boundaries between coarse blocks emerge directly from the dynamics of human-guided motion. This makes the procedure no more labor-intensive than collecting a single ordinary demonstration.

We explored alternative strategies for fully automatic segmentation, such as keyframe/keypose detection and curvature-based change-point identification James et al. (2022); Shridhar et al. (2023); Ma et al. (2024). However, reliably identifying semantically meaningful waypoints from a single demonstration proved brittle, especially when waypoints must correspond to subtle but crucial phases of contact establishment, re-grasping, or dual-arm synchronization. Hardware-assisted approaches (e.g., instrumented gloves as in DexCap Wang et al. (2024a) or end-effector teaching interfaces as in UMI Chi et al. (2024) and DexUMI Xu et al. (2025)) can offer additional cues for segmentation but introduce alignment overhead, cross-morphology calibration challenges, and potential loss of semantic correspondence between human motion and robot embodiment. Given these trade-offs, the kinesthetic-teaching-driven segmentation remains a pragmatic and robust choice: it imposes no additional annotation burden, preserves semantic alignment across blocks, and provides boundaries well suited for constructing invariant and variant components for real-world synthesis.

## B Details of Data Collection, Processing and Synthesis

### B.1 Task-Oriented Collection Standards

To systematically study factors influencing policies, we establish task-specific data collection protocols that ensure spatial uniformity and instance balance. Each task defines a common workspace (blue quadrilateral in Fig. 10, mapped to a 615mm×675mm rectangular area on the table) where objects are placed. For single-object tasks (*e.g.*, `unscrew`), objects are positioned across grid cells (white dots/boxes in Fig. 10) covering the entire workspace. For dual-object tasks, the workspace is split into left-right regions (*e.g.*, bottles occupy the left half, mugs the right half in `pouring`), with objects distributed uniformly within their zones. Object orientations are randomized for `plugpen`, `inserting`, and `reorient` but fixed for `unscrew`, `pouring`, and `pressing` (see Fig. 1 in the main paper for orientation counts). A grid cell is marked as "collected" if its centroid hosts a demonstration, allowing local variations with positional tolerance. This protocol balances coverage and practicality, enabling controlled studies on spatial and instance-level generalization.

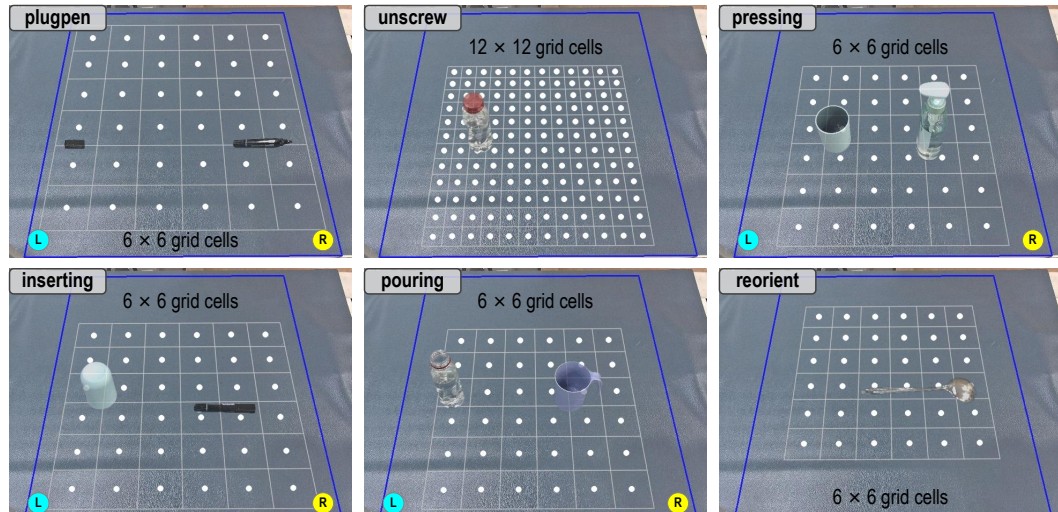

Figure 10: The specific grid cell division way for each task. For tasks involving two manipulated objects, the total number of grid cells will be divided equally between the left side and right side.

## B.2 COLLECTION UI INTERFACE DEVELOPMENT

To streamline data collection, we develop a real-time UI interface (Fig. 11) that visualizes task-specific grid cells, dynamically tracks object positions/poses, and validates alignment with the one-shot demonstration. The interface overlays grid layouts onto live camera feeds, highlights detected objects (using YOLOE Wang et al. (2025) for fast detection and segmentation), and marks completed cells using red crosses. The position of objects is represented by red dots (some objects also use yellow arrows to indicate their orientation). In addition, for cups and bottles that do not need to pay attention to changes in orientation, we did not use the mask centroid as their representative point during collection, but instead used the lowest endpoint in contact with the desktop. A sidebar displays positional/orientational deviations from the reference demonstration, enabling quick verification. If object perception fails, the system falls back to Florence2 Xiao et al. (2024) and SAM2 Ravi et al. (2024) for robust detection and segmentation. Operators trigger a "capture-align-validate-save" cycle every 2 seconds. And successful saves prompt visual feedback (UI flash of corresponding grid cells), signaling readiness for the next sample. This setup eliminates manual grid marking and minimizes human idle time, achieving a high speedup while ensuring spatial consistency. Please refer our **Supplemental Videos** that demonstrate the vivid workflow, illustrating how the UI balances efficiency with precision.

## B.3 POST-PROCESSING OF COLLECTED DATA

This part includes the last two stages of BiDemoSyn: initial frame alignment and trajectory optimization (defined in the method part of main paper). First, pre-recorded positional/orientational deviations (validated via the UI interface during collection) align novel observations with the one-shot demonstration reference frame using the rigid transformation. Next, trajectory optimization ensures kinematic feasibility. We prune singular configurations and validate collision-free paths using the robot IK solvers and motion planners. Over 96% of trajectories pass validation, with failures (*e.g.*, unreachable corner grid cells) either re-collected or discarded. Finally, validated trajectories are stored in $\mathcal{D}_{syn}$ to complete the synthesis pipeline.

## B.4 CLARIFICATION ON HYPERPARAMETERS AND PREDEFINED SYMBOLS

In Sec. 4, several task-specific thresholds and predefined parameters are introduced for clarity and reproducibility. Here we further explain their meaning and selection strategy. For instance, threshold symbols $\delta$, $\zeta$ and $\gamma$ in Eqn. 4, Eqn. 5 and Eqn. 6 are predefined only for one-shot deconstruction. The symbol $\delta$ is a minimal motion saliency threshold, used to identify boundaries between coarse blocks. The symbol $\zeta$ is a static tolerance threshold to segment motion boundaries under quasi-static assumptions. The symbol $\gamma$ is a refinement threshold used to create fine-grained AEPs after bounding block duration ($T_{\max}$). And the scaling factor $\lambda$ in Eqn. 10 controls the adaptation from

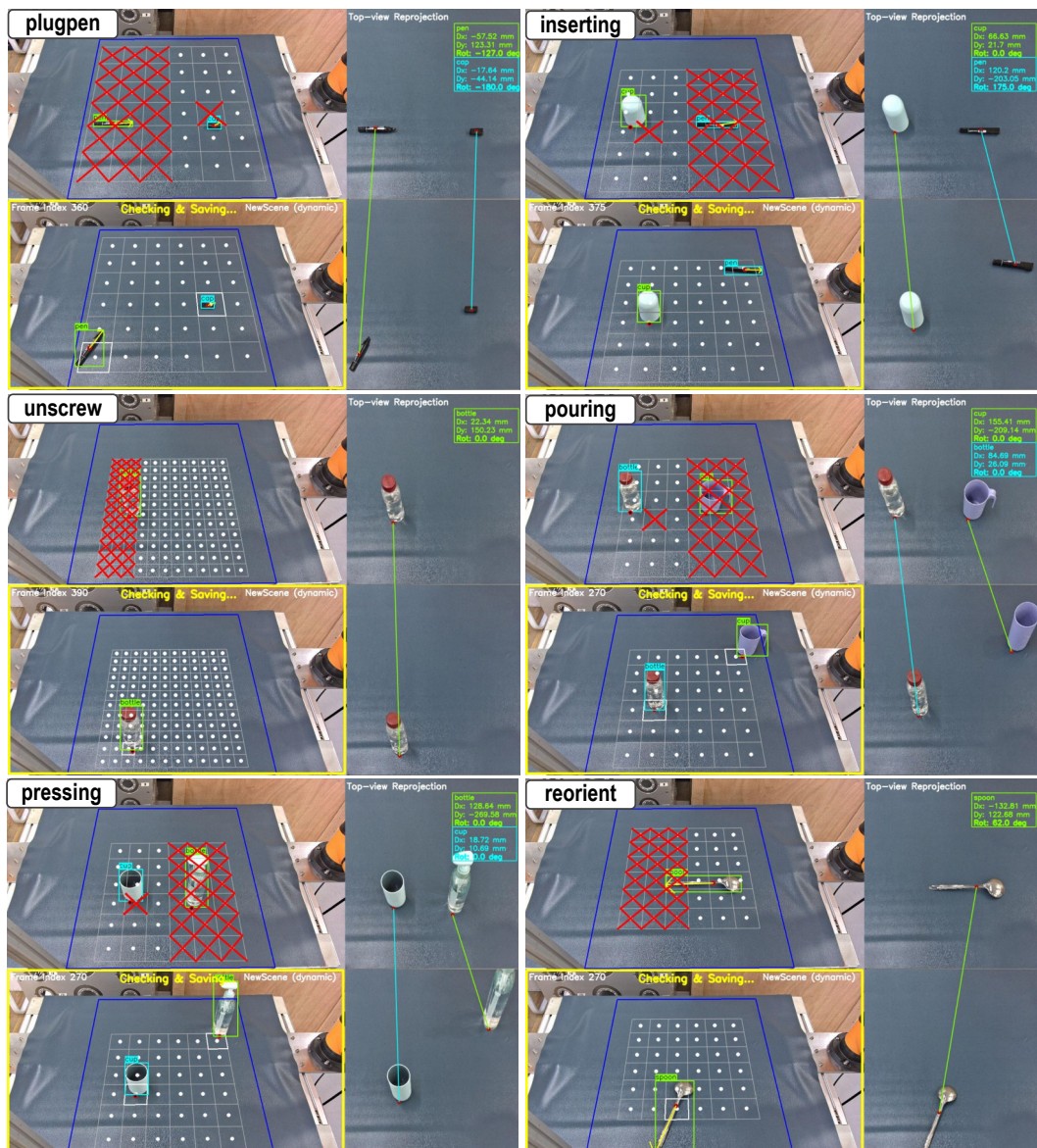

Figure 11: UI interface examples. When collecting data, the six tasks need to display different grid cells in real time, as well as perceive and track objects related to the tasks. Note that these points, crosses and lines are drawn digitally, which are not marks in the real world.

the canonical one-shot demonstration to new object instances with varying sizes. Theoretically, $\lambda \in \mathbb{R}^+$, yet empirical tuning shows that values within $[0.8, 1.0]$ are sufficient for stable grasp transfer without destabilizing contact. Larger or smaller values tend to shift grasp points away from functional regions (e.g., bottle mid-section in unscrew), while $\lambda = 1.0$ already suffices for most tasks. Similarly, thresholds for collision-checking or pose-alignment rejection (e.g., rejecting infeasible IK solutions or near-singular configurations) were determined through small-scale grid search and validated in pilot runs. Importantly, such hyperparameters are not per-trial tunable knobs, but rather fixed system constants, ensuring consistent data synthesis across tasks.

## B.5 ADDITIONAL EVALUATION OF INTERMEDIATE PROCESSES

We provide further evaluation to justify intermediate design choices from three aspects: Segmentation Quality, Pose Estimation Robustness and Impact on Data Reliability. (1) Firstly, object masks were obtained from Florence2 Xiao et al. (2024) and SAM2 Ravi et al. (2024), two state-of-the-art vision foundation models (VFMs). On held-out captures without foreground occlusion, the segmen-

tation accuracy exceeded 97%, ensuring reliable geometric inputs for subsequent processing. (2) Secondly, the axis alignment for objects with a distinct orientation was based on classical image-moment theory: first-order moments define centroids, second-order moments define orientation. This method, given reliable masks, yields highly stable results. Quantitative inspection of $\sim$200 sampled trials revealed $<$2% pose estimation errors, and only under severe occlusion. (3) Thirdly, in Sec. D.2 (**Statistics and Analysis of Failed Cases**), we showed that perception and orientation errors together account for 47% of failures, highlighting their importance. Nevertheless, these inter-mediate modules are already close to optimal under current open-loop constraints, which explains the high success rate ( 98%) of the data collection system.

### B.6 Positioning with Respect to Prior Work

To avoid potential confusion with prior works such as ReKep Huang et al. (2024) and ODIL Wang & Johns (2025), we emphasize both the similarities and the fundamental differences. Like these methods, BiDemoSyn exploits one-shot demonstrations and compositional reasoning; however, *its core objective is not direct policy execution*, but rather *rapid and scalable synthesis of real-world training demonstrations*. This distinction is crucial: while prior methods target zero-/few-shot policy generalization (often trading reliability for flexibility), BiDemoSyn deliberately focuses on max-imizing data quality and efficiency, enabling downstream training of robust visuomotor policies (e.g., DP3 Ze et al. (2024) and EquiBot Yang et al. (2024a)). Our evaluation against DemoGen Xue et al. (2025) and YOTO Zhou et al. (2025), which are two strong baselines addressing the same bottleneck, demonstrates that BiDemoSyn achieves a superior balance between efficiency and data reliability, which is the practical need for scaling imitation learning.

A concurrent work MoMaGen Li et al. (2025) tackles scalable bimanual mobile manipulation by generating diverse trajectories in *simulation*, focusing on base reachability and camera visibility through constrained optimization. Although sharing a high-level goal of reducing human effort, MoMaGen and BiDemoSyn operate under fundamentally different assumptions: MoMaGen re-lies on noise-free simulation assets and virtual scene perturbations, whereas BiDemoSyn performs *real-world demonstration synthesis*, directly addressing physical-domain challenges such as noisy perception, contact feasibility, and one-shot decomposition under real sensor observations. As a result, MoMaGen expands diversity through simulated scene variation, while BiDemoSyn enables *physically grounded generalization* across spatial and category-level object variations without sim-to-real transfer. Thus, BiDemoSyn should be viewed not as a direct competitor to simulation-based frameworks like MoMaGen, but as a *complementary real-world solution* specifically designed for scalable, high-fidelity data generation where simulation pipelines cannot fully capture real-world physical constraints.

### B.7 Evaluation of Demonstration Quality

Evaluating the quality of synthesized demonstrations is essential because the downstream visuo-motor policies depend on both the fidelity of observations and the physical feasibility of actions. We assess demonstration quality along two orthogonal dimensions that align with the structure of visuomotor imitation learning:

**Visual Fidelity**. This dimension evaluates whether the observations included in a synthesized demonstration faithfully reflect the real physical scene without introducing artifacts. Teleoperation and BiDemoSyn preserve raw RGB-D observations, ensuring perfect consistency with the physical environment. YOTO Zhou et al. (2025) and DemoGen Xue et al. (2025) modify the seed point cloud to reposition objects, which introduces geometric distortions and depth inconsistencies, particularly as the displacement grows (see Fig. 4B). These artifacts degrade the policy's perception module by coupling unrealistic geometry with real sensor noise.

**Trajectory Feasibility and Physical Grounding**. A demonstration is considered high-quality if the associated action sequence: (1) yields a collision-free, dynamically stable trajectory, (2) respects the contact geometry implied by the task semantics, and (3) can be executed on the real robot without additional correction or replanning. Teleoperation and YOTO Zhou et al. (2025) provide ground-truth trajectories. BiDemoSyn produces partially reused (invariant blocks) and partially adapted (variant blocks) trajectories aligned with real-world object pose and geometry. DemoGen Xue et al.

(2025) synthesizes trajectories directly from edited scene representations, which can deviate from physically stable contact geometries.

Finally, Fig. 4A reflects the interplay of these two components. Demonstrations with high-quality RGB-D observations and physically feasible actions rank highest. This ordering also predicts downstream policy performance: policies trained with cleaner, physically grounded demonstrations demonstrate higher success rates and better robustness under OOD evaluation. These expanded analyses clarify the qualitative and quantitative factors driving the reported comparisons and support the role of demonstration quality as a key determinant of real-world imitation learning performance.

### B.8 Diversity of Synthetic Demonstrations

A natural question is whether additional diversity (such as reordering or randomizing block sequences) would improve the effectiveness of the synthesized dataset. In BiDemoSyn, we intentionally do not introduce such sequence-level variations. Our empirical findings align with recent analyses in large-scale teleoperation datasets focusing on the diversity exploration and ablation for scalable robotic manipulation Shi et al. (2025), which show that low-level behavioral variability (e.g., minor differences in grasp location, end-effector speed profiles, or redundant micro-motions) does not necessarily benefit policy learning and may even hinder convergence. Instead, we adopt the principle that *task success is primarily determined by achieving the correct functional interactions, not by how those interactions are executed.*

Accordingly, in the design of BiDemoSyn, we maintain the original causal structure of the task by preserving the block ordering extracted from the one-shot demonstration, while introducing diversity along the *dimensions that matter for generalization*, namely: (1) spatial variation of object pose (translation + rotation), and (2) category-level instance variation (shape and size differences within the same object class). These forms of variation are directly tied to visuomotor generalization and produce clear benefits in downstream policy performance (see Tab. 1). In contrast, randomizing the internal block structure introduces behavioral diversity that is orthogonal to task semantics and does not improve success rates in our preliminary tests. We therefore avoid unnecessary variation and instead focus on *environment-induced diversity*, which is both meaningful and consistent with the underlying task geometry.

## C Training and Deployment of Imitation Policies

### C.1 Reproduction of Baselines

For DemoGen Xue et al. (2025), we synthesize trajectories by editing the 3D point cloud in one-shot demonstration: objects are translated across grid cells (aligned with Sec. B.1 standards) and rotated (for orientation-sensitive tasks) to match our BiDemoSyn spatial diversity for fair comparison. Edited point clouds exhibit artifacts (*e.g.*, misaligned edges, self-occluded regions) due to perspective distortions, degrading observation fidelity. For YOTO Zhou et al. (2025), we reduce data scale to 1/10 of BiDemoSyn to accommodate its time-consuming physical replay. Even at this scale, YOTO requires 388/777/115/259/129/100 replays across all six tasks (*e.g.*, `plugpen` needs 388 replays, taking about 11 hours), versus BiDemoSyn's 3-hour synthesis for $10\times$ data. Note that the efficiency recorded here is the time consumed by continuous collection, including a lot of non-collection time for transition and error recovery. The spatial sampling of YOTO is relaxed to coarse left-right object placements, sacrificing grid uniformity.

### C.2 Implementation of Imitation Policies

Both DP3 Ze et al. (2024) and EquiBot Yang et al. (2024a) (refer Fig. 12) ingest segmented 3D object point clouds but differ in network architectures. Specifically, DP3 employs specially designed multiple MLPs to encode point clouds into latent features, then directly regressing dual-arm keyposes. EquiBot enhances robustness via a SIM(3)-equivariant PointNet variant: point clouds undergo rotation/scale-invariant feature extraction before action prediction. This equips EquiBot with improved generalization to unseen object variations (refer Tab. 1 in the main paper), achieving 5–12% higher success rates than DP3 under identical training data and settings. Both policies oper-

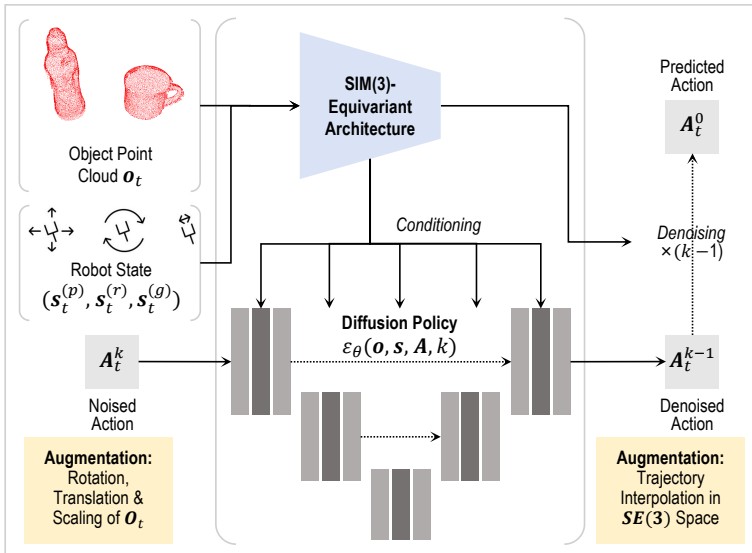

Figure 12: The adapted visuomotor policy network architecture for imitation learning of bimanual manipulation tasks. The input visual observation is simplified to a point cloud of manipulated objects. Note that the vision encoder is slightly different in DP3 Ze et al. (2024) and EquiBot Yang et al. (2024a). The illustrated one is for EquiBot.

ate in open-loop, aligning with synthesized demonstration formats. Below, we give a more detailed description of the visuomotor policy modification for the dual-arm action prediction.

- **Spaces of observation and action**. We adopt a 13-dimensional proprioception vector and a 7-dimensional action space for each robot arm, respectively. The proprioception data for each arm consists of the following information: a 3-dimensional end-effector position, a 6-dimensional vector denoting end-effector orientation (represented by two columns of the end-effector rotation matrix), a 3-dimensional vector indicating the direction of gravity, and a scalar that represents the degree to which the gripper is opened. The action space for each arm consists of the following information: a 3-dimensional vector for the end-effector position offset, a 3-dimensional vector for the end-effector angular velocity in axis-angle format, and a scalar denoting the gripper action. For all bimanual tasks, the observation horizon is set to 1, so we only use the initial state observation of left arm as one of the network inputs. And the initial state of right arm is always fixed in each task. For the number of action steps (also the length of the predicted horizon), we simplify it and set the prediction length to the number of keyposes $K$, which can be extracted from the start and end actions of each Atomic Execution Primitives (AEPs) during task deconstruction.

- **Network architecture**. For DP3, it is a variant of Diffusion Policy Chi et al. (2023) with a simpler point cloud encoder. It also designs a two-layer MLP to encode robot proprioceptive states before concatenating with the observation representation. For EquiBot, we use a SIM(3)-equivariant PointNet++ Yang et al. (2024b) with 4 layers and hidden dimensionality 128 as the feature encoder. For the noise prediction network, we inherits hyperparameters from the original Diffusion Policy. Specifically, to optimize for inference speed in all experiments, we use the DDIM scheduler Song et al. (2021) with 8 denoising steps, instead of the DDPM scheduler Ho et al. (2020) which performs up to 100 denoising steps.

- **Sampling of point cloud**. As we all known, setting the number of points to sample in the point cloud observation is a key hyperparameter to consider when designing an architecture that takes point cloud inputs. In our experiments, we found out that using 1024 points is sufficient for all tasks and policies. In particular, we have tried increasing the number of point clouds to 2048 or more, but the evaluation improvement in each task is minimal, and this will also cause the storage occupied by the training observation data to be too large and the training time cost to increase. Therefore, reducing the number of points to 1024 can make training faster without hurting performance. And all our policy models can be trained on a GeForce RTX 3090 Ti with 24 GB of memory.

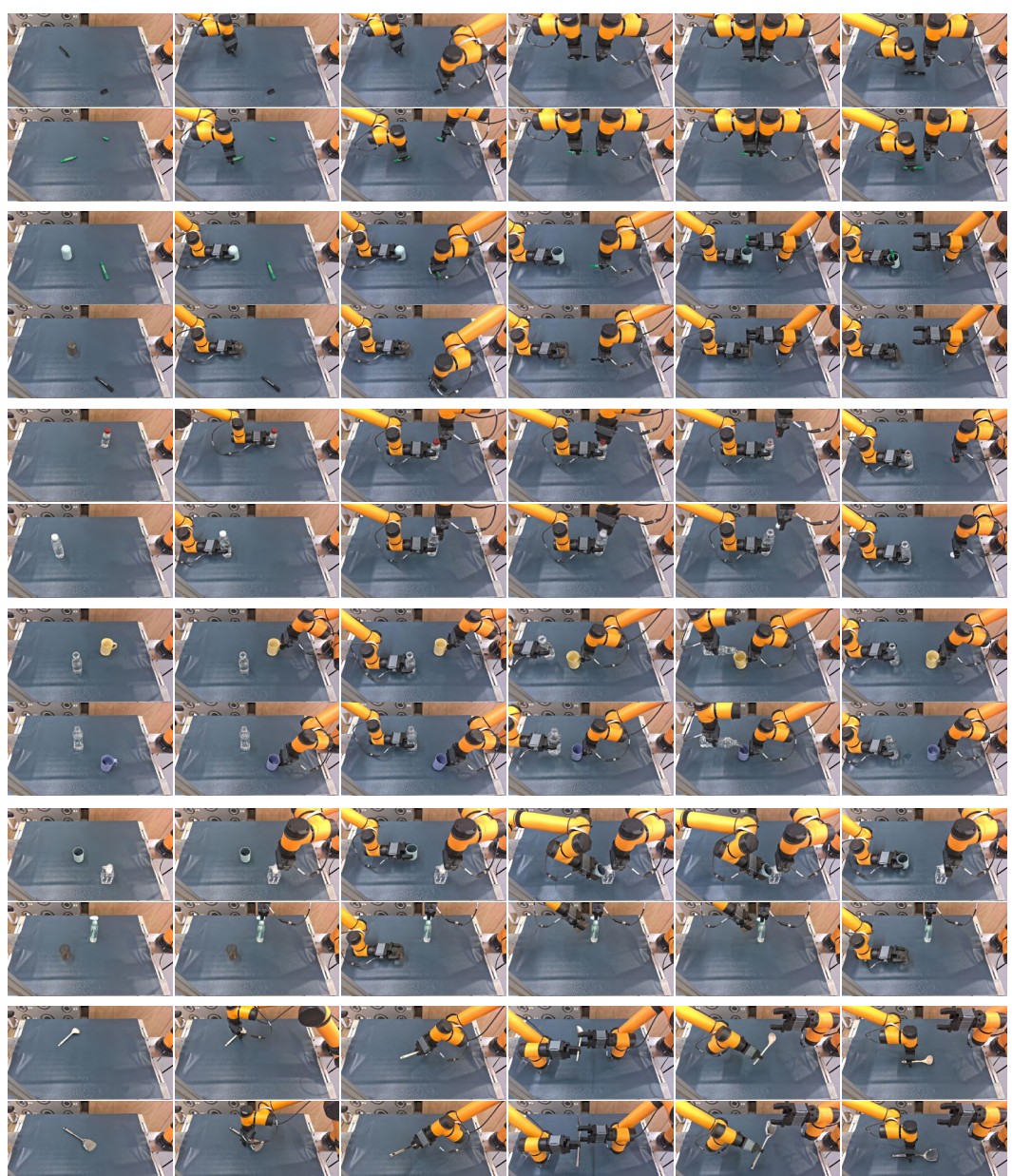

Figure 13: Qualitative real rollout samples from the left RGB camera for all evaluation scenarios. From top to bottom, they are defined six bimanual tasks. Best to view after zooming in.

- **Training and evaluation**. When conducting the in-distribution (ID) and out-of-distribution (OOD) evaluations with full demonstrations, we train all methods for 500 and 1,000 epochs, respectively. Otherwise, when using the 1/10 training data (especially the YOTO), we train all methods of the ID and OOD evaluations for 2,000 and 4,000 epochs, respectively. For all experiments, the batch-size is always set to 64. We only evaluate the last one checkpoint saved at the end of training. For every evaluation in the real world, we run the policy in a randomly initialized placement of objects, and record the average success rate achieved by the policy.

In addition, we have adopted the object-centric point cloud input. At inference time, we also need to preprocess the binocular RGB observations to obtain the point cloud of manipulated objects. This core design relies on the still rapidly developing capabilities of vision foundation models (VFMs). Here we leverage SOTA open vocabulary detection method Florence-2 Xiao et al. (2024) and segmentation method SAM2 Ravi et al. (2024) to automatically extract object masks and then filter out corresponding point clouds. Despite this, occasionally we may fail to segment desired objects accu-

rately, and in these special cases we will manually correct the masks. These cases are not counted as failed evaluation trails due to not involving significant elements of bimanual robot manipulation. Because we believe that the next generation of VFMs can alleviate these problems, or we can directly address them through domain adaptation, test-time adaptation, or adjusting input prompts.

# D   MORE REAL ROBOT RESULTS VISUALIZATION AND ANALYSIS

## D.1   MORE EXAMPLES FROM REAL ROLLOUTS

As shown in Fig. 13, it supplements more qualitative results with additional real-robot execution visualizations, highlighting nuanced state transitions and task-specific challenges. These visualizations underscore BiDemoSyn's ability to handle real-world complexities (such as mechanical tolerances and imperfect perception), while also exposing limitations in dynamic force modulation (*e.g.*, over-pressing bottles or lids). Please refer our **Supplemental Videos** which provide frame-by-frame analyses of these executions, further dissecting critical phase and offering insights into policy decision-making under uncertainty.

## D.2   STATISTICS AND ANALYSIS OF FAILED CASES

While policies trained with BiDemoSyn achieve high success rates, failure analysis reveals systematic challenges. Note that although the policy we trained is executed end-to-end (it establishes an implicit mapping from observations to action outputs), we can still align the analysis from the stage where its failure cases are located to find the core cause of the error. Finally, using the experimental results of EquiBot under out-of-distribution evaluations, we categorize failures into five types according to the task execution logic (mainly from the design module of BiDemoSyn):

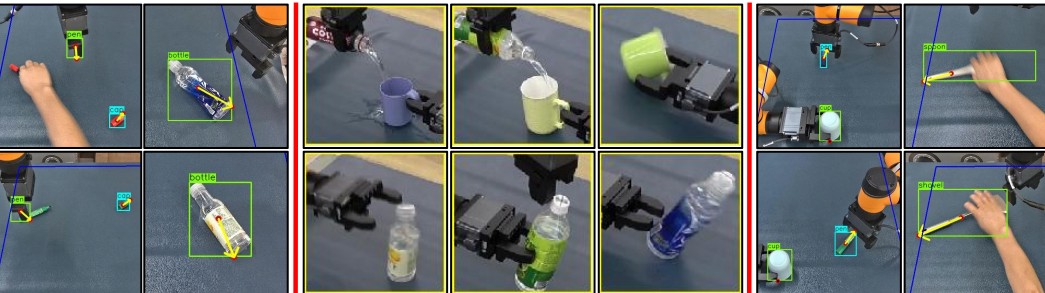

Figure 14: Typical failure cases. **Left four cases:** These include two cases from the `plugpen` task that incorrectly detected the marker body, and two cases from the `reorient+unscrew` task that incorrectly estimated the lying bottle's orientation. **Middle six cases:** These include three cases from the `pouring` task where water was spilled or the mug cup was misgrabbed, and three cases from the `unscrew` task where the gripper failed to grab the bottle's body or cap accurately (these are extra footages from a third-person perspective). **Right four cases:** These include two cases in the `inserting` task where the robotic arm's end effector occasionally obscures the marker pen, and two cases in the `reorient` task where human interference causes the hand to partially obscure the spoon/shovel. Here, visual perceptual defects do not necessarily lead to task failure, demonstrating the robustness of the proposed orientation estimation algorithm.

- **Visual Perception Errors (15%)**: Inaccurate object segmentation (*e.g.*, mistaking the tabletop as part of the marker pen) leads to misaligned grasps.

- **Orientation Estimation Errors (32%)**: Incorrect object pose estimation (*e.g.*, misjudging a spoon's concave direction) causes malformed trajectories (*e.g.*, flipping to the wrong side).

- **Alignment Failures (18%)**: Vision-guided initial frame misalignment (*e.g.*, bottle nozzle offset by >1cm) propagates errors to subsequent steps (*e.g.*, spilling during `pouring`).

- **Initial Grasp Failures (28%)**: Collisions during pre-grasp motions (*e.g.*, gripper nudging objects) or unstable grasps (*e.g.*, slippage in `reorient`) prevent task initiation.

- **Error Propagation (7%)**: Cumulative errors from earlier stages compound into irreversible failures (*e.g.*, slightly misaligned pluging between pen body and pen cap in `plugpen`).

As can be seen, the orientation estimation and initial grasp failures dominate, reflecting two core challenges: (1) current pose estimators struggle with symmetric or textureless objects (*e.g.*, metal spoon or shovel), and (2) gripper-centric path planning lacks fine-grained contact modeling (*e.g.*, avoiding pre-touch collisions for irregular shapes). Addressing these requires advances in category-agnostic pose estimation and short-horizon contact optimization, which are critical directions for our future work. Some visual failure examples are shown in Fig. 14, which provides a foundation for a full understanding of the BiDemoSyn system and its robust adaptation to allow for dynamic pre-grasping. More dynamic rollouts can be found in our **Supplementary Videos**.

### D.3   TWO MORE COMPLEX, LONG-HORIZON TASKS

To further evaluate the scalability of BiDemoSyn beyond the six primary bimanual tasks, we introduce two additional long-horizon tasks that require sequential manipulation steps and accumulate significantly more physical error throughout the execution:

- `reorient+unscrew`: The robot must first upright a lying down bottle (reorient phase), then perform a dual-arm unscrewing motion to remove the cap (unscrew phase). This task stresses both precise estimation of bottle orientation and robust reuse of multi-step, contact-rich interaction patterns.
- `unscrew+pouring`: The fisrt phaset is the original unscrew task. After unscrewing the bottle cap, the robot must lift a mug cup, perform dual-arm coordination to tilt the bottle, and pour water into the cup. This composition integrates two high-precision subtasks and requires stable arm coordination across multiple transitions.

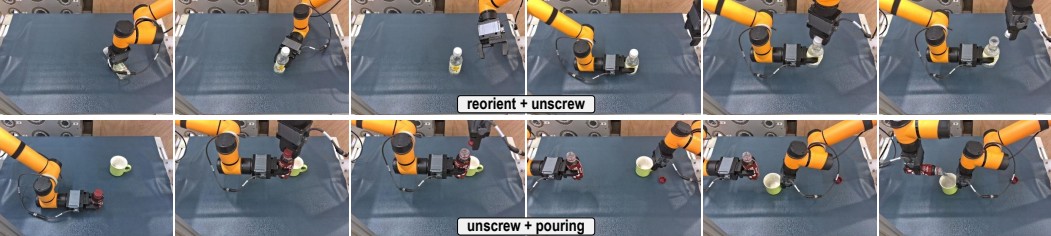

Figure 15: Qualitative real rollout samples from the left RGB camera for two newly added complex long-horizon bimanual tasks. Best to view after zooming in.

**Data Synthesis Using BiDemoSyn**.  These two newly added tasks were synthesized following the same three-stage pipeline described in the main text. We briefly summarize the configurations: for `reorient+unscrew`, we collected initial observations for 4 bottle instances, each placed at 7 orientations across 36 grid cells, yielding **1008 synthesized demonstrations**; for `unscrew+pouring`, we collected initial observations for 2 bottle and 2 cup instances, each placed on 18 grid cells, yielding **1296 synthesized demonstrations**. Importantly, both tasks introduce *longer temporal horizons, sequential coordination, and increased compounding error*, thus providing a stronger stress test than the original six tasks.

**Policy Training and Performance**. Using the synthesized datasets, we trained visuomotor policies for each long-horizon task based on the adapted EquiBot architecture (the strongest performer in Tab. 1 of the main text). Across both tasks, the trained policies exhibited high robustness to spatial variation (different placements and bottle orientations), reliable execution of multi-step dual-arm coordination, and consistent task success under mild external perturbations introduced during pre-grasp interpolation. Fig. 15 presents some qualitative results with additional real-robot execution visualizations of these two new tasks. And Fig. 16 provides illustrative examples of continuous dynamic interference scenarios among tasks `pouring` and `reorient+unscrew`. Full dynamic demonstrations and rollouts of all eight tasks (previous 6 tasks and the new 2 tasks) are included in the **Supplemental Videos**. These results further validate BiDemoSyn's ability to rapidly scale to new, long-horizon tasks with minimal additional engineering, and reinforce its suitability as a practical framework for synthesizing large-scale, real-world training data.

### D.4   FEW-SHOT-BASED DEMONSTRATION SYNTHESIS

While BiDemoSyn is designed around a one-shot paradigm, the framework naturally extends to a few-shot setting, where multiple real demonstrations may serve as complementary seeds. This

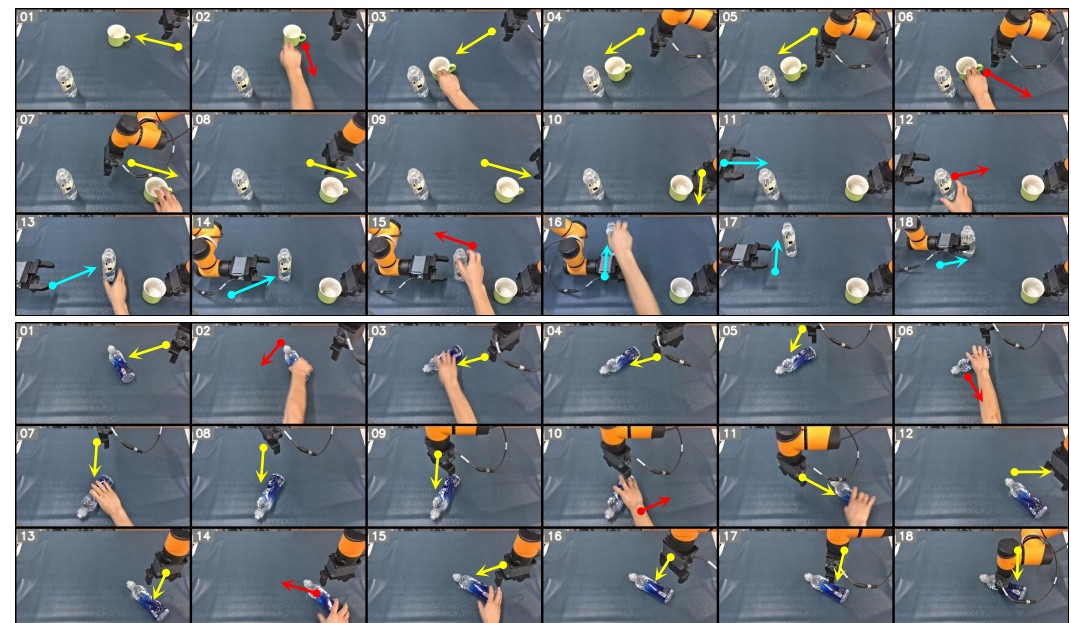

Figure 16: Qualitative examples of applying dynamic interferences during the pre-grasping execution. From top to bottom, they are segments of the dynamic closed-loop grasping phase of tasks `pouring` and `reorient+unscrew`, where each object is manually disturbed from one to three times. The red arrow indicates the direction of the manually moved object (interfering). The cyan arrow and yellow arrow indicate the movement direction of the left and right robotic arms (chasing) respectively. Best to view after zooming in.

Table 2: Quantitative comparison results of success rates for the few-shot BiDemoSyn under *in-distribution* and *out-of-distribution* evaluations. The # means the BiDemoSyn training data size. Although the number of demonstrations used for training is always the same for each setting, the number of seed demonstrations used for trajectory synthesis is different.

| Trained Policy | Few-Shot BiDemoSyn | *in-distribution* evaluations | | | | | | | *out-of-distribution* evaluations | | | | | | |
|---|---|---|---|---|---|---|---|---|---|---|---|---|---|---|---|
| | | plugpen | inserting | unscrew | pouring | pressing | reorient | Average Success Rate | plugpen | inserting | unscrew | pouring | pressing | reorient | Average Success Rate |
| | | #3888 | #7776 | #1152 | #2592 | #1296 | #1008 | | #2916 | #3888 | #1008 | #972 | #324 | #756 | |
| EquiBot | 1-shot | 28/30 | 28/30 | 26/30 | 25/30 | 24/30 | 25/30 | 86.7% | 18/30 | 20/30 | 24/30 | 21/30 | 17/30 | 20/30 | 66.7% |
| | 5-shot | 28/30 | 28/30 | 26/30 | 26/30 | 25/30 | 25/30 | 87.8% | 20/30 | 21/30 | 24/30 | 23/30 | 18/30 | 21/30 | 70.0% |
| | 10-shot | 28/30 | 28/30 | 26/30 | 27/30 | 25/30 | 26/30 | 88.9% | 21/30 | 21/30 | 25/30 | 23/30 | 18/30 | 22/30 | 71.7% |
| | 15-shot | 28/30 | 28/30 | 27/30 | 27/30 | 26/30 | 26/30 | 90.0% | 21/30 | 22/30 | 25/30 | 23/30 | 18/30 | 22/30 | 72.2% |
| | 20-shot | 28/30 | 28/30 | 27/30 | 27/30 | 26/30 | 26/30 | 90.0% | 21/30 | 22/30 | 25/30 | 23/30 | 18/30 | 22/30 | 72.2% |

subsection studies the effect of incorporating a small number of additional demonstrations (5–20 samples) obtained through standard teleoperation or kinesthetic teaching (used in this study).

**Few-Shot Integration into BiDemoSyn**. Given multiple seed demonstrations, we incorporate them in two ways: (1) *Enhanced Object Variation Coverage*. Additional demonstrations expand the diversity of observed object geometries, grasp points, and intermediate contact poses. These are incorporated into the variant blocks during trajectory adaptation. (2) *Improved Robustness of Reference Motion*. Multiple demonstrations allow BiDemoSyn to aggregate or select more reliable canonical motion patterns for the invariant blocks, mitigating single-demonstration idiosyncrasies. Under these two settings, the followed overall synthesis pipeline remains unchanged.

**Experimental Findings**. We have summarized results of this ablation study in Tab. 2. These new experiments show: (1) *In-distribution (ID) performance*. Adding few-shot data provides little to no improvement. Policies trained with BiDemoSyn's one-shot–generated datasets already exhibit near-saturation performance for seen objects and placements. (2) *Out-of-distribution (OOD) performance*. Few-shot data provides moderate gains, primarily by increasing the number of observed object instances and geometric variations, thereby improving generalization to unseen shapes and dimensions. These findings align with the intuition that BiDemoSyn already produces high-quality,

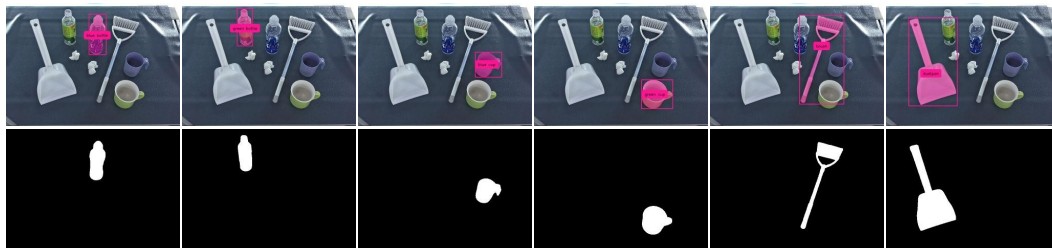

Figure 17: **Top row**: Example cluttered tabletop scenes used to evaluate the perception front-end of BiDemoSyn. Each scene includes the target objects for the `pouring` task—two bottles (text prompts are *blue bottle* and *green bottle*) and two mugs (text prompts *blue cup* and *green cup*)—surrounded by visually distracting household items. Two additional tool-like distractors (*brush*, *dustpan*) are intentionally introduced to support future extensions such as dual-arm `sweeping` tasks. **Bottom row**: High-quality 2D object masks produced by Florence-2 + SAM2 for all queried categories. Despite clutter, color similarity, and semantic ambiguity, the VFM stack consistently isolates each target object with precise boundaries, providing reliable anchors for subsequent pose inference and variant-block adaptation in BiDemoSyn.

physically faithful demonstrations from a single demonstration (refer Fig. 5). Additional few-shot demonstrations mainly help by expanding the representational coverage rather than correcting failure cases caused by hardware limitations of the embodied robot entity itself.

**Additional Conclusion**. The few-shot extension confirms that BiDemoSyn supports scalable integration of multiple demonstrations, maintains strong performance in the one-shot setting, benefits from few-shot seeds mostly in the OOD regime, and requires no modification to its core synthesis pipeline. This suggests a promising avenue for future research: exploring hybrid one-shot + few-shot paradigms that balance data-collection cost with maximal generalization capability.

## D.5 VFM-BASED OBJECT ANCHORING IN CLUTTER

To evaluate the applicability of BiDemoSyn under more realistic and visually complex environments, we additionally examine its object-anchoring capability when the target objects are placed in cluttered scenes with distractors, varied background patterns, and partial occlusions. Although the main experiments intentionally isolate environmental variables to study the core challenge of real-world demonstration synthesis, BiDemoSyn is fully compatible with more complex perception conditions due to its reliance on Vision Foundation Models (VFMs) for object segmentation.

**Clutter-Aware Object Localization**. We employ Florence-2 Xiao et al. (2024) and SAM2 Ravi et al. (2024) to detect and segment the task-relevant items (e.g., bottle, mug) using either text queries or exemplar-based prompts. In cluttered tabletop scenes, the VFM stack consistently produces high-quality masks, even when distractors share similar colors or shapes. Qualitatively, the predicted masks remain stable across moderate occlusions, enabling the system to isolate the target object with minimal ambiguity. Fig. 17 illustrates representative examples where both bottle and mug are correctly segmented despite surrounding tools, utensils, packaging, and nonrigid paper balls.

**Implications for Demonstration Synthesis**. Because BiDemoSyn's variant-block adaptation depends only on (1) reliable 2D masks and (2) centroid- and principal-axis–based orientation inference, accurate object anchoring is the only requirement for extending the system to cluttered environments. Our experiments show that the VFM-based perception pipeline provides sufficiently robust segmentation to support (a) pose-aligned trajectory modulation and (b) collision-aware grasp adjustments, even when the background is visually nonuniform.

**Limitations and Future Directions**. While VFMs significantly enhance robustness to visual complexity, extreme occlusions or heavy inter-object entanglement may still degrade performance. Extending BiDemoSyn to handle these cases will require multi-view sensing, active view planning, or integrating VLM-driven object-query disambiguation. Nevertheless, these initial results demonstrate that *VFM-based anchoring provides a practical and reliable foundation for deploying BiDemoSyn beyond clean tabletop settings*, thereby enabling future extensions to household, warehouse, and mobile manipulation scenarios.

## E  SUMMARY, REFLECTION AND OUTLOOK

This work presents BiDemoSyn, a framework that bridges the reality gap in bimanual imitation learning by synthesizing diverse, physically feasible demonstrations from a single human example. Through systematic decomposition, vision-guided alignment, and hierarchical optimization, our method eliminates simulation dependencies while scaling data collection by orders of magnitude. Experiments across six contact-rich tasks validate its efficiency (5 seconds per demo), scalability, and generalization to unseen spatial and geometric variations.

However, limitations persist: BiDemoSyn currently assumes quasi-static motions, struggles with highly dynamic tasks (*e.g.*, catching), and faces challenges in extreme shape variations (*e.g.*, deformable objects). Failures primarily stem from orientation estimation errors and grasp-time collisions, highlighting needs for category-agnostic pose estimators and contact-aware motion planners. Future work will extend BiDemoSyn to deformable object manipulation, integrate dynamic perception for moving targets, and explore hybrid learning paradigms (*e.g.*, combining synthetic demos with reinforcement learning). By democratizing real-world demonstration synthesis, this work aims to accelerate progress toward generalizable robotic manipulation.

## F  USE OF LARGE LANGUAGE MODELS

In preparing this manuscript, we employed the GPT-4o language model *strictly for text editing purposes*, such as improving readability and polishing grammar. The model was not used to generate research ideas, design methodologies, conduct experiments, or analyze results. All substantive contributions remain entirely those of the authors.

