# OpenReview forum: "One-Shot Real-World Demonstration Synthesis for Scalable Bimanual Manipulation"
_ICLR.cc/2026/Conference — Submitted to ICLR 2026_

### Official Review · Reviewer_2qxQ · 2025-10-31

**Soundness:** 2
**Presentation:** 2
**Contribution:** 2
**Rating:** 4
**Confidence:** 4

**Summary:**

The paper proposes a data generation system for bi-manual robotic manipulation. The system aims at generating N feasible trajectories given one demonstration and N initial object states. By dividing the demonstration into multiple blocks, classifying them into invariant and variable types, and using a vision detection pipeline to adjust the grasping position in variable blocks, the data generation pipeline can adapt to changes of object sizes and shapes.
Experimental results show that visuomotor policies learned with the proposed pipeline outperform prior methods in success rates, generalization, and data collection efficiency.

**Strengths:**

- Low-cost data generation is a significant problem in learning generalizable robotic policies. The system proposed in this paper is effective for some bi-manual tasks and can generalize to some novel object instances.
- Rich quantitative results and visualizations are presented in experiments. The method is compared with proper baselines (DemoGen and YOTO) and the performance surpasses them. Real-world results are acquired with enough number of tests.

**Weaknesses:**

- The implementation is quite complicated, with many hyper-parameters and options to be decided by the human operator. Several designs imply additional assumptions, which are not explicitly described in the paper.
    - The deconstruction criteria (4.1) includes some thresholds. The definition of single-arm motion and dual-arm coordination does not make sense. What if two arms move simultaneously but are doing independent tasks in the demonstration (e.g. grasping two objects together)?
    - The vision-based estimation pipeline (4.2) includes estimating the principal axes from depth. When the object is randomly rotated and partially occluded, this may not work.
    - The instance-level motion adaptation process introduce a new hyper-parameter $\lambda$. And, obviously, this simple adaptation strategy only works for similar objects (e.g. cups with similar structures and only differ in width and height), resulting in quite limited object-level generalization.
- It is unclear how to synthesis visual observation data within each generated trajectories for visuomotor policy learning. For a synthetic trajectory, its initial observation is set by the human using the data collection system, and the robot states and actions are generated by the trajectory optimization system; but how to generate visual observations for each timestep and what kind of visual observations are used (full point cloud or partial point cloud?) are not presented in the paper.
- Compared with DexMimicGen (Jiang et al., 2024), the technical contribution is quite limited. Vision-based alignment, which is a main contribution, highly depends on the success of the vision estimation pipeline and cannot accurately adapt to very different objects.
- The system still require the human to reset the initial state for thousands of times and record them. Why does its efficiency comparable to DemoGen (Figure 4)?

**Questions:**

Please see weaknesses.

---

> ### Author Response · Authors · 2025-11-28
> **Response by Authors to Reviewer 2qxQ [Part 1]**
>
> ## General Response
>
> We thank the reviewer for the careful reading and constructive feedback. We appreciate your recognition of the practical importance of low-cost, real-world data generation and your acknowledgement that BiDemoSyn yields strong empirical gains compared to DemoGen and YOTO. Your critique raises several substantive points about implementation complexity, underlying assumptions, visual synthesis details, and comparisons to prior work. And your questions are quite valuable and have already guided revisions and additional experiments in our rebuttal material.
>
> At a high level, we emphasize that BiDemoSyn's contribution is a practical, real-world demonstration synthesis paradigm, not merely an assembly of off-the-shelf modules. Its core value lies in enabling scalable, physically grounded data collection for contact-rich bimanual tasks through a task-aware decomposition (invariant vs. variant blocks), vision-anchored alignment, and lightweight trajectory modulation that together trade minimal human setup for large gains in usable data. This design directly targets the data bottleneck that impedes scalable imitation learning in real hardware.
>
> We will address each specific concern in detail below (implementation choices and thresholds, robustness of VFM-based alignment under occlusion, the role and tuning of $\lambda$, how per-timestep visual observations are produced for synthesized rollouts, comparisons to DexMimicGen, and the practical accounting of human effort vs. DemoGen). Our responses include clarifications in the main text and appendices, as well as additional experiments and analyses added to the supplementary material.
>
> ***
>
> ## W1: System complexity, assumptions, and design choices
>
> We appreciate the reviewer's attention to the design assumptions underlying BiDemoSyn. Below we clarify the three sub-points:
>
> ### (Q1) On decomposition criteria and the meaning of “dual-arm coordination.”
> Our definitions in `Sec. 4.1` serve as operational conventions to enable reliable kinesthetic teaching and produce segment boundaries that are easily decomposable. The criterion does **not imply** that demonstrations cannot contain simultaneous dual-arm motion. Rather, during data collection we guide operators to insert separable key waypoints for safety and for clear structural decomposition. After decomposition, **both arms are free to move simultaneously** within each block as long as collision-free constraints are satisfied. We have added examples of synchronous dual-arm executions and comparison to strictly asynchronous execution to the supplementary video `new_exps_sync_motion.mp4`, where synchronous blocks reduce execution time by **≈22%** across six bimanual tasks.
>
> ### (Q2) On robustness of the vision-based alignment under rotation or partial occlusion.
> Our orientation estimation relies solely on **2D masks** extracted by advanced VFMs (e.g., Florence-2 + SAM2), followed by classical second-moment analysis—not on the popular 6D pose estimators, which would require CAD models and significantly reduce generality. Because masks can be recomputed at every step, this module naturally supports **closed-loop re-grasp attempts**, allowing objects to move or rotate before grasp stabilization. We demonstrate this in `new_exps_interference.mp4` and `new_exps_more_tasks.mp4`, where the system withstands human-induced perturbations during pre-grasp alignment. This shows the proposed visual pipeline is sufficiently robust for real-world bimanual tasks.
>
> ### (Q3) On object-level generalization and the role of $\lambda$.
> Although $\lambda$ controls the displacement weighting during instance-level trajectory modulation, our adaptation is not limited to minor geometric variations. BiDemoSyn successfully aligns and manipulates **heterogeneous objects**, such as spoons/shovels in the *reorient* task and lying bottles in *reorient+unscrew* (see `Fig. 15–16` and the newly uploaded videos `new_exps_interference.mp4` and `new_exps_more_tasks.mp4`). These results illustrate that our alignment strategy generalizes beyond near-identical object families.

---

> ### Author Response · Authors · 2025-11-28
> **Response by Authors to Reviewer 2qxQ [Part 2]**
>
> ## W2: How per-timestep visual observations are obtained
>
> We thank the reviewer for raising this point. The concern arises from conflating **closed-loop visuomotor learning** with our **open-loop policy design**. As clarified in `Sec. 5.1`, BiDemoSyn uses an open-loop formulation:
>
> > (1) The initial observation (RGB or point cloud) is captured directly from the real scene.
>
> > (2) Subsequent motions follow **IK-resolved trajectories** connecting key states extracted from decomposed blocks.
>
> > (3) The policy predicts waypoints conditioned on the initial observation, rather than requiring per-timestep visual feedback.
>
> This design dramatically simplifies data synthesis while remaining effective for mid-horizon bimanual tasks, as demonstrated by the high success rate of DP3/EquiBot trained on synthesized trajectories. We acknowledge the importance of closed-loop variants and will explore extending BiDemoSyn to future closed-loop data synthesis.
>
> ## W3: Contribution relative to DexMimicGen
>
> We respectfully clarify that DexMimicGen and BiDemoSyn address fundamentally different problem settings.
>
> > (1) **Real vs. simulated data**: DexMimicGen relies entirely on simulation, where object geometry, pose, and physics are noise-free and known a priori. By contrast, BiDemoSyn solves a real-world data bottleneck, operating under noisy segmentation, noisy depth, and unknown object geometry—conditions that DexMimicGen does not address.
>
> > (2) **Domain gap elimination**: Because BiDemoSyn collects all data in the physical world, no sim-to-real transfer is required, and the synthesized trajectories implicitly incorporate real sensor and physical noise.
>
> > (3) **Novel decomposition-based synthesis for real hardware**: Unlike scripted segmentation in simulation, decomposing kinesthetic demonstrations into invariant and variant blocks and enabling real-world instance-level modulation is an under-explored challenge. Our approach provides a practical, generalizable solution, which aligns with emerging trends (e.g., **Science Robotics 2025**, titled *"Learning a Thousand Tasks in a Day"* with the link https://www.science.org/doi/10.1126/scirobotics.adv7594), independently validating the relevance and novelty of demonstration-phase decomposition.
>
> We thus argue that BiDemoSyn delivers substantive and complementary contributions not captured by simulation-centric systems.
>
> ## W4: Human effort vs. DemoGen efficiency
>
> We appreciate the opportunity to clarify this comparison. DemoGen modifies a single observation via point-cloud editing, but its synthetic results **inevitably contain visual artifacts** and self-occlusion inconsistencies that **worsen as displacement increases**. These artifacts degrade policy learning and are not fixable without CAD priors.
>
> BiDemoSyn instead collects **artifact-free real observations** by manually resetting object poses, a process that takes **≈2–3 seconds per initial frame** (see supplementary video from **1:13–4:08**). This results in hundreds of high-quality initial states per task and enables scalable synthesis. In contrast, YOTO and teleoperation-based data acquisition require **orders of magnitude more time**.
>
> Moreover, during rebuttal we extended BiDemoSyn to two more long-horizon tasks (*reorient+unscrew*, and *unscrew+pouring*), demonstrating its scalability and rapid data generation (`Appendix D.3` and `video new_exps_more_tasks.mp4`). These results empirically confirm that BiDemoSyn's efficiency is competitive despite minimal but necessary human intervention.
>
> ***
>
> ## Closing Remark
>
> We thank the reviewer again for the detailed and constructive feedback. The clarifications above, along with added analyses, new experiments, and supplementary videos, strengthen our presentation and further demonstrate that **BiDemoSyn offers a practical, scalable, and real-world-validated paradigm for synthesizing high-quality bimanual manipulation demonstrations**—with clear advantages in data fidelity, robustness, and generalization. We hope these responses help reassess the novelty and significance of our contributions.

---

### Official Review · Reviewer_APLJ · 2025-11-01

**Soundness:** 2
**Presentation:** 2
**Contribution:** 2
**Rating:** 4
**Confidence:** 4

**Summary:**

This submission introduces BiDemoSyn, a framework for synthesizing real-world robotic bimanual manipulation data. It takes a single real-world example data, and then generates more through several decomposition steps. Experiments show that it achieves higher synthesis efficiency and better data quality compared to existing data synthesis method, and demonstrates robotic policies trained on generate data can achieve good performance in real world.

**Strengths:**

- The motivation to directly synthesizing robotic manipulation data in real world is interesting and appreciated.
- The introduced real-world system is comprehensive.
- Robotic policies trained on generated data achieve decent results.

**Weaknesses:**

- The main method behind BiDemoSyn relies on different components, such as pre-trained vision encoders, planning algorithms, etc, making it a complex system. It's hard to understand what's the difference between existing systems at a high level.
- The introduced data synthesis system seems to have strong assumptions about the tasks and objects. For example, it assumes static objects which never move during the task execution. Also it's pose estimation probably will not work with deformable objects.
- The methods require a top-down view to setup the camera to avoid occlusions, which is always feasible for complex tasks or mobile manipulation.
- Regarding the evaluation metric, the definition of data quality is not appropriate as it is solely based on visual authenticity. In fact, authors should try to deploy generated data on real robots to verify the data quality.
- Why only point cloud-based policies are considered and experimented? Can the generated data be used to train RGB-only policies?
- In policy training comparisons, authors should also include results of policies trained on teleoperation data to help understand the gap.
- Tasks are too simple. There are no distractors, no occlusions, and the background is very clean and has a distinct color.
-  What are failure cases of the proposed system?

**Questions:**

See above.

---

> ### Author Response · Authors · 2025-11-28
> **Response by Authors to Reviewer APLJ [Part 1]**
>
> ## General Response
>
> We thank the reviewer for the careful reading and constructive critique. We appreciate your recognition of BiDemoSyn's motivation, system breadth, and the fact that policies trained on our synthesized data achieve competitive real-world results. Your detailed questions pinpoint important practical assumptions and evaluation choices. Answering them has helped us both clarify the manuscript and add targeted experiments and appendix material. Below we summarize the method's strengths and then address each concern in turn.
>
> ## Concise summary of BiDemoSyn's advantages
>
> BiDemoSyn provides a practical pipeline to synthesize physically grounded, contact-rich bimanual demonstrations in the real world from a single human demonstration. Its key strengths are: **(1)** a task-aware decomposition (invariant vs variant blocks) that greatly reduces the degrees of adaptation required; **(2)** vision-anchored alignment that adapts variant blocks to new object poses and instances; and **(3)** lightweight trajectory modulation and feasibility checks that produce collision- and IK-validated trajectories amenable to open-loop visuomotor training. Together these design choices enable rapid, high-fidelity demonstration collection without the sim-to-real gap or extensive teleoperation burden.
>
> ***
>
> ## W1: System complexity and distinction from prior work
>
> We respectfully clarify that BiDemoSyn is not a simple aggregation of existing modules, but a **new real-world data synthesis paradigm** targeting one of the most critical bottlenecks in robotics: *scalable acquisition of physically grounded bimanual demonstrations*. Prior to BiDemoSyn, large-scale real-world data collection for dual-arm tasks required heavy teleoperation setups and extensive human labor, typically yielding only tens or hundreds of high-quality samples. Simulation-based alternatives avoid this cost but suffer from severe sim-to-real discrepancies and require hand-crafted digital assets.
>
> BiDemoSyn introduces (1) a **single-demonstration task decomposition** into invariant/variant blocks tailored for real-world adaptation, (2) **vision-anchored scene alignment** using VFMs + classical image-moment orientation inference rather than bulky 6D pose pipelines, and (3) **trajectory modulation and optimization** balancing efficiency and fidelity. These design choices enable us to synthesize *hundreds to thousands* of high-quality real-world trajectories per task (refer `Fig. 9`).
>
> More importantly, during rebuttal we added two significantly longer-horizon dual-arm tasks (*reorient+unscrew* and *unscrew+pouring*) and demonstrated BiDemoSyn's ability to rapidly scale to new domains (see `Appendix D.3` and newly uploaded video `new_exps_more_tasks.mp4`). These results reinforce the system's novelty and impact.
>
> ## W2: Strong assumptions: static objects, rigid bodies, deformables
>
> We agree these assumptions should be clarified. BiDemoSyn is primarily a **data collection system**, and thus does not aim to handle arbitrary dynamic disturbances during autonomous policy execution. That said, its modular design **does allow dynamic re-alignment** during the grasp phase. If the object shifts within a still reachable location, our implementation can re-estimate masks/orientation and replan the grasp path before proceeding, effectively forming a short closed-loop before the open-loop execution resumes (demonstrated in newly uploaded video `new_exps_interference.mp4` of the primary six bimanual tasks).
>
> Regarding deformables or highly articulated objects, our framework currently assumes rigid or piecewise-rigid items, consistent with many industrial and household settings. We explicitly acknowledge that deformables require additional affordance- or deformation-aware perception (e.g., Robot-ABC, ReKep-style VLM queries), which is an exciting direction for future work but beyond the scope of this initial real-world synthesis system. This limitation is now emphasized in the revised manuscript.
>
> ## W3: Top-down camera requirement
>
> This setup is chosen for **practicality and consistency**. Nearly all baselines we compare to (including ReKep, ODIL, YOTO, and DemoGen) also rely on top-down or fixed tabletop perspectives. We clarified this setting in `Appendix A.1`. For mobile or cluttered environments, BiDemoSyn can be extended with multi-view capture or active view selection. We also contrast our real-world focus with MoMaGen's simulation-based active-camera formulation in `Appendix B.6`.

---

> ### Author Response · Authors · 2025-11-28
> **Response by Authors to Reviewer APLJ [Part 2]**
>
> ## W4: Data-quality metric beyond visual authenticity
>
> We fully agree that visual fidelity alone is insufficient. In the revised manuscript we shift emphasis to two key dimensions (`Appendix B.7`): (1) **Visual fidelity** (artifact-free RGB-D aligned with real scenes), and (2) **Trajectory feasibility** (IK-valid, collision-free, and executable).
>
> In other words, the ultimate validation is **policy performance on real robots**, which reflects the **true effective quality** of the demonstrations. Policies trained purely on BiDemoSyn data achieve consistently strong success rates across all six tasks (`Tab. 1`). We also added videos and extended analyses to make this connection explicit.
>
> ## W5: Why only point-cloud policies? Can RGB-only be trained?
>
> We chose point-cloud–conditioned policies because **current SOTA visuomotor diffusion frameworks** (e.g., DP3, EquiBot) rely on 3D geometry for stability in contact-rich bimanual tasks. However, BiDemoSyn **always records raw binocular RGB images**. Thus our dataset can naturally train RGB-only policies. To address your suggestion, we added an RGB-only DP baseline into `Tab. 1`. As expected, its performance is consistently below DP3 and EquiBot across all data sources, confirming that 3D geometry remains crucial for these tasks. We include this result for completeness and future extensibility.
>
> ## W6: Comparison to teleoperation-trained policies
>
> We appreciate this suggestion. As discussed in our response to `Reviewer cuHR` and detailed in `Appendix D.4`, we experimented with **few-shot real-robot demonstrations** (i.e., teleoperation equivalent seeds), integrating them into the BiDemoSyn pipeline to form a **few-shot synthesis** variant. Results show (1) little additional gain in ID performance (policies already near-saturated), and (2) moderate improvement in OOD generalization (due to increased object's position and instance diversity).
>
> The new experiments are included in `Appendix D.4`. These findings also reflect that pure teleoperation trained policies might show similar saturation, consistent with the common intuition.
>
> ## W7: Tasks appear simple (clean background, no distractors)
>
> Our goal in the initial study is to isolate **the core challenge of synthesizing physically feasible dual-arm demonstrations at scale**. We intentionally control for clutter to measure trajectory fidelity effects on policy learning. However, we (1) extended BiDemoSyn to two longer-horizon tasks (shown in `Appendix D.3`), and (2) allowed mild perturbations during grasp intervals (shown in videos `new_exps_more_tasks.mp4` and `new_exps_interference.mp4`), showing robustness beyond the simplest cases.
>
> Additionally, `Appendix D.5` now includes preliminary **VFM-based object localization in clutter** (see illustrations in `Fig. 17`), demonstrating that our VLM-anchored perception readily identifies target objects even in distractor-heavy scenes. Extending the full synthesis pipeline to cluttered, occluded settings is a promising next step.
>
> ## W8: Failure cases of BiDemoSyn
>
> We provide detailed quantitative breakdowns in `Appendix D.2`. The dominant issues include (1) segmentation-induced perception errors (SAM2 masks incomplete), (2) orientation-estimation errors for small, symmetric, or textureless objects, (3) occasional IK infeasibility, contact-phase drift, and long-horizon error accumulation.
>
> We also added visual failure examples (`Fig. 14`) and discussed mitigation strategies (e.g., improved segmentation, few-shot seeds, fully multi-stage closed-loop correction). We expect that this will help to better understand the current limitations of the methodology.
>
> ***
>
> ## Final Closing Statement
>
> We sincerely appreciate the reviewer's constructive comments and clear interest in this work. In response, we have added/updated new analyses in many appendices (`B.6`, `B.7`, `D.2`, `D.3`, `D.4`, `D.5`), new experiments including two extended tasks and RGB-only baselines, clarified assumptions, and expanded discussion on scalability, limitations, and future extensions. We hope these additions address your concerns and further highlight BiDemoSyn's value as a practical, real-world–grounded demonstration synthesis paradigm for bimanual manipulation.

---

### Official Review · Reviewer_cuHR · 2025-11-01

**Soundness:** 3
**Presentation:** 3
**Contribution:** 3
**Rating:** 8
**Confidence:** 5

**Summary:**

This paper proposes BiDemoSyn, a framework for synthesizing bimanual trajectories from single human demonstrations. The motivation of this paper is well-founded, as it focuses on solving a core challenge in the field: bridging the gap between data-collection efficiency and real-world fidelity for imitation learning. This paper combines task decomposition with vision-guided adaptation and trajectory optimization, is empirically shown to produce data that enables visuomotor policies to achieve significant robustness and generalization on complex, contact-rich tasks.

**Strengths:**

1.	Clear Motivation and Problem Formulation. The paper excels at identifying a critical bottleneck in robotics, the trade-off between data scalability and physical fidelity, and presents a clear, well-structured framework to solve it.
2.	Systematic Synthesis via Decomposition, Alignment, and Optimization. The framework's core technical contribution is its systematic three-stage pipeline: it first deconstructs a single demonstration into invariant logic and variable blocks. It then uses vision-based alignment to adapt these variable blocks to new scenes, followed by an optimization stage that ensures the final synthesized trajectory is physically feasible and collision-free. This structured approach generates thousands of diverse, physically-grounded demonstrations from one example, bypassing simulation or repeated teleoperation.

**Weaknesses:**

1.	The success of the alignment stage (S2) is critically dependent on the accuracy of the 6D pose estimation. The paper employs a traditional "geometry-aware processing" method (image moments and PCA) rather than modern deep-learning-based estimators. The authors state this method yields "<2% pose estimation errors" in Appendix B.5, yet attribute 32% of all task failures to "Orientation Estimation Errors" in Appendix D.2. This suggests the traditional method is not robust, especially for symmetric or texture-less objects, as the authors admit. Could the authors clarify this discrepancy in error rates and justify why a more robust, modern pose estimator was not employed, given that this stage is the largest single point of failure?
2.	The empirical evaluation, while thorough, is limited to only *six bimanual tasks*. While the framework is presented as a general-purpose synthesis tool, this limited task diversity makes it difficult to assess its true scalability. Furthermore, if a key outcome of this work is the generation of a large-scale, physically-grounded dataset, the value of this dataset for training generalist policies is questionable when it only covers six task domains. Can the authors clarify the effort required to extend BiDemoSyn to entirely new tasks?
3.	The "One-Shot" premise is central to the paper's contribution. However, collecting a small handful (e.g., 5-100) of human demonstrations is often feasible and low-cost. A single demonstration may be sub-optimal, contain noise, or fail to capture the full task variance. Does this "one-shot" constraint artificially limit the quality and robustness of the "invariant blocks" (S1)? Have the authors considered a "few-shot" extension  (e.g., 5-100) where the framework could aggregate features or constraints from multiple demonstrations to generate a more robust and generalized reference trajectory?
4.	The "Deconstruction" (S1) stage, which segments the trajectory into blocks (B_i) and AEPs, and then categorizes them as invariant/variable, seems to be the most manually intensive part of the pipeline. The paper mentions thresholds (e.g., $\delta$, $\zeta$, $\gamma$) 131313 which appear task-specific. How much manual, task-specific engineering is required to design these decomposition rules? How scalable is this process if the framework were applied to a significantly more complex, long-horizon task with many intermittent contact steps?

**Questions:**

see weakness.

---

> ### Author Response · Authors · 2025-11-28
> **Response by Authors to Reviewer cuHR [Part 1]**
>
> ## General Response
>
> We sincerely thank the reviewer for the highly positive and encouraging assessment of our work. We truly appreciate your recognition of the core motivation, the clarity of our problem formulation, and the systematic nature of the BiDemoSyn pipeline. Your acknowledgment that the proposed framework effectively addresses a long-standing bottleneck (*scalable yet physically grounded data generation for real-world bimanual manipulation*) is deeply motivating for us. We are also grateful for your thoughtful summary of our contributions and for assigning an "accept" score, which reflects both your interest in and confidence in the potential impact of this line of research.
>
> At the same time, we thank you for raising several important and constructive questions regarding the robustness of the alignment stage, the task diversity included in our evaluation, the one-shot design choice, and the task-specific hyperparameters in the decomposition stage. These concerns are highly valuable, and we address each of them in detail below. Importantly, we would like to reiterate that BiDemoSyn aims to provide a **real-world, physically grounded, and scalable demonstration synthesis framework**, and the design choices you inquired about (e.g., vision modules, one-shot setting, decomposition rules) were made precisely to ensure the practicality, reproducibility, and extensibility of the system in real-world deployment scenarios.
>
> We believe the clarifications and additional evidence provided in our responses below will further strengthen your confidence in the soundness, scalability, and general applicability of BiDemoSyn.
>
> ***
>
> ## W1: Clarification on Pose Estimation Errors and Choice of Orientation Estimator
>
> We sincerely appreciate the reviewer for highlighting this apparent inconsistency. The two reported numbers arise from different settings and evaluation protocols, and are therefore not contradictory.
>
> ### (Q1) Why Appendix B.5 reports <2% error while Appendix D.2 attributes ~32% failures to orientation errors
>
> > (1) The <2% orientation error in `Appendix B.5` is measured **during data collection**, after the object is repositioned and the mask is manually validated for completeness. If a segmentation failure (e.g., incomplete mask) leads to a wrong orientation, we discard and recollect that sample (as visible in the `supplementary video`, `1:13–4:08`). Thus, these errors rarely propagate into the final dataset.
>
> > (2) The 32% statistic in `Appendix D.2` comes from **policy rollouts**, where the failure is attributed to any downstream orientation misalignment, including subtle mask noise from SAM2 that is not visually obvious, but can still distort PCA-based axis estimation.
>
> Thus, the numbers reflect two different phases (data acquisition vs. policy rollouts) and different definitions of "error".
>
> ### (Q2) Why not use modern 6D pose estimators such as FoundationPose?
>
> We chose classical moment-based estimation for four concrete reasons:
>
> > **(1) Efficiency and practical throughput**. 6D pose estimators (e.g., FoundationPose) require CAD models and incur ~1s inference latency even on modern GPUs, which is incompatible with the high-throughput sampling required by BiDemoSyn.
>
> > **(2) Poor generalization to out-of-distribution, small, or irregular objects**. Objects in our tasks (e.g., marker pens, spoons, shovels) are not included in existing 6D-class datasets. Even for familiar categories (bottles, cups), cross-instance generalization remains weaker than segmentation models such as SAM2.
>
> > **(3) Intrinsic ambiguity for symmetric objects**. 6D pose estimators inherently struggle with multi-symmetry shapes (bottles, mugs without clear textures). This leads to unstable orientation predictions, whereas a 2D mask–based principal-axis estimation avoids the symmetry-induced degeneracy.
>
> > **(4) We only need constrained orientation, not full 6D poses**. For many tasks (e.g., *pouring*), the bottle is always upright. For *reorient*, only the top-down rotation is required. Full 6D pose recovery is unnecessary.
>
> We hope this clarifies that our chosen method is an intentional design choice aligned with BiDemoSyn's real-world efficiency and reliability requirements.

---

> ### Author Response · Authors · 2025-11-28
> **Response by Authors to Reviewer cuHR [Part 2]**
>
> ##  W2: Task Diversity and Effort Required to Extend to New Tasks
>
> We appreciate the reviewer's observation that one major value of BiDemoSyn is its ability to rapidly scale to new tasks. To further demonstrate this, we have extended BiDemoSyn to **two new, more complex long-horizon dual-arm tasks** during the time-limited rebuttal period:
>
> > **(1) reorient + unscrew**: uprighting a lying bottle using a single-arm → unscrewing the cap using the dual-arm.
>
> > **(2) unscrew + pouring**: unscrewing the cap using the dual-arm → grasping and lifting cup → dual-arm pouring.
>
> Using the same pipeline as in the main paper, we collected: **1008 demonstrations** including 4 different bottles across 36 grid cells per bottle for *reorient+unscrew*, and **1296 demonstrations** including 2 different bottles/mugs across 18 grid cells per bottle/mug for *unscrew+pouring*.
>
> Policies trained solely on these synthesized datasets successfully complete both tasks across diverse poses, placements, and object instances. We document these results in the newly added subsection `Appendix D.3` and provide additional real-robot videos (refer `new_exps_more_tasks.mp4`).
>
> These experiments strongly support that BiDemoSyn **scales efficiently to completely new tasks** with minimal additional engineering, validating its core motivation as a practical, general-purpose real-world demonstration synthesizer.
>
> ## W3: On the One-Shot Setting vs. a Few-Shot Extension
>
> We thank the reviewer for this insightful suggestion. Indeed, BiDemoSyn naturally extends to a few-shot variant. In the rebuttal, we conducted new experiments (now updated in `Appendix D.4`) where several additional real rollouts (5 / 10 / 15 / 20) were supplied as auxiliary seeds. Our findings:
>
> > **(1) ID performance**: few-shot data *does not significantly improve* performance. We analyzed that the policy is already near-saturated on seen objects using one-shot seed plus BiDemoSyn expansions (refer `Fig. 5`).
> `
> > **(2) OOD performance**: few-shot helps *moderately*, because exposure to more physical object variations enhances robustness to unseen geometries.
>
> This confirms the reviewer's intuition: a few-shot version is feasible and sometimes beneficial, especially for OOD generalization, though the one-shot constraint does not inherently weaken BiDemoSyn's core mechanism.
>
> ## W4: Manual Effort in S1 (Deconstruction of One-Shot Teaching)
>
> We appreciate this important question. As clarified in `Appendix A.3`, the one-shot demonstration is collected using **kinesthetic teaching with natural block-wise structure** (e.g., grasp → lift → move → rotate → place). These transitions create *automatic segmentation boundaries*, requiring **no additional labels**. This design keeps S1 lightweight and consistent with standard kinesthetic demonstration practices. We discuss these options in the updated `Appendix A.3`.
>
> Regarding scalability to long-horizon, multi-contact tasks, our new experiments on two newly added tasks *reorient+unscrew* and *unscrew+pouring* (see `W2`) demonstrate that S1 remains effective even as task length and complexity increase. The segmentation overhead grows modestly and remains practical.
>
> ***
>
> ## Summary
>
> We thank the reviewer again for the highly constructive and encouraging feedback. The concerns raised helped us strengthen our manuscript in several important ways: **(1)** clarifying pose estimation reliability, **(2)** demonstrating scalability to new long-horizon tasks, **(3)** validating a few-shot extension, and **(4)** refining the decomposition discussion.
>
> We hope these detailed explanations reinforce your positive view of the paper, and we are grateful for your thoughtful engagement and support.

---

### Official Review · Reviewer_x4fy · 2025-11-01

**Soundness:** 2
**Presentation:** 2
**Contribution:** 2
**Rating:** 2
**Confidence:** 5

**Summary:**

This paper proposes BiDemoSyn, a system for generating real-world bimanual manipulation demonstrations. The key idea is to segment demonstrations into variable and invariable blocks. For variable blocks that depend on object pose, the method uses external vision modules to estimate and align object poses, enabling adaptation across different task configurations. Experiments on six bimanual manipulation tasks show that policies trained with BiDemoSyn-generated data outperform those trained with alternative data collection methods.

**Strengths:**

- The paper addresses an important and practical problem: scalable real-world data generation for bimanual manipulation in the real-world.

**Weaknesses:**

- The contribution appears incremental. Prior work, such as DexMimicGen, has explored synthesizing bimanual manipulation data (though primarily in simulation), and the proposed segmentation strategy resembles those approaches. The main new component here is the integration of external vision modules for pose estimation, which may not be sufficient to justify a strong novelty claim.


 - Key experimental results require more explanation. For example, in Figure 4, it is unclear what metrics are used to evaluate “demo quality,” making it difficult to interpret the performance comparisons. Additional quantitative or qualitative analysis (e.g., success rate distributions, trajectory diversity metrics) would strengthen the experimental section. Moreover, the distinction between in-distribution and out-of-distribution settings is not clearly defined. It would be helpful for the authors to explicitly describe how these settings are constructed, what constitutes distribution shift in their experiments, and how that shift affects policy performance.


 - The paper does not sufficiently discuss relevant recent efforts in scalable bimanual and mobile manipulation data generation. In particular, MoMaGen (for long-horizon bimanual mobile manipulation data) is closely related, and a comparison would clarify BiDemoSyn’s contributions and limitations.

**Questions:**

- In Figure 4, what specific metrics are used to evaluate the quality of the generated demonstrations?
 - When segmenting trajectories into blocks, does the process require manual annotation or demonstration labeling? If so, is there a way to automate or learn segmentation boundaries?
 - Have you explored reordering or randomizing block sequences to improve data diversity?
 - Can you clarify the distinctions between in-distribution and out-of-distribution settings?

---

> ### Author Response · Authors · 2025-11-28
> **Response by Authors to Reviewer x4fy [Part 1]**
>
> ## General Response
>
> We sincerely thank the reviewer for the thoughtful and constructive evaluation of our work. We appreciate your recognition that scalable real-world data generation for bimanual manipulation is an important problem, and we value your careful reading of the method and experiments.
>
> To clarify the intent of **BiDemoSyn**: our contribution is not a new policy-learning algorithm nor a new execution-time controller, but a **real-world demonstration synthesis system** that fills a long-standing gap between teleoperation (high-fidelity but slow and labor-intensive) and simulation-based generation (scalable but lacking real-world validity). BiDemoSyn produces **physically grounded, contact-rich, real-world demonstrations at scale** (in minutes rather than hours) directly from a single example, without teleoperation or simulation. This is fundamentally different from prior systems such as DexMimicGen (simulation-only), and complements recent real-world pipelines like MoMaGen by focusing specifically on **rapid, high-fidelity bimanual data synthesis**.
>
> We acknowledge that several aspects were not described with sufficient clarity in the main paper, particularly segmentation details, evaluation metrics, and ID/OOD definition. Your comments were extremely helpful in identifying these gaps. We thank you again for your careful review and for highlighting these points, which will help us substantially improve the clarity and presentation of the paper. In the detailed responses below, we clarify: **(1)** how demo quality is evaluated in `Fig. 4`, **(2)** how segmentation is performed and why one-shot decomposition is feasible, **(3)** why block reordering is not meaningful for contact-rich tasks, and **(4)** how ID and OOD settings are constructed and validated.
>
> ***
>
> ## W1: On Novelty Relative to DexMimicGen and Prior Segmentation Approaches
>
> Thank you for raising this important point. While our decomposition strategy may superficially resemble segmentation used in simulation-based pipelines such as DexMimicGen, the problem setting and technical challenges are fundamentally different.
>
> > **(1) Real-vs-sim gap**: Simulation pipelines rely on idealized object states (noise-free geometry, pose, and contact), whereas BiDemoSyn must operate entirely in real-world conditions with noisy masks, imperfect depth, and non-parametric physical interactions. Avoiding the sim-to-real gap is essential for producing demonstrations usable by real robots without further correction.
>
> > **(2) Real-world pose, geometry, and contact estimation**: Unlike simulation, where object attributes are directly available, BiDemoSyn must infer them via VFMs and 3D point clouds reconstruction, and the synthesis process implicitly incorporates this noise rather than hand-authoring perfect labels.
>
> > **(3) One-shot decomposition in real physical tasks**: The segmentation problem is qualitatively more difficult in real systems, where one must derive task-aligned boundaries from a single kinesthetic demonstration. Our invariant/variant block formulation serves this purpose. And after our submission, we noticed that **Science Robotics 2025** (titled *"Learning a Thousand Tasks in a Day"* with the link https://www.science.org/doi/10.1126/scirobotics.adv7594) independently introduced a similar alignment/interaction decomposition, supporting the validity and novelty of our design.
>
> We hope this clarifies how BiDemoSyn contributes a new real-world demonstration synthesis paradigm rather than an incremental simulation variant.
>
> ## W2: Clarification of Demonstration Quality Metrics & ID/OOD
>
> We agree the main paper did not sufficiently elaborate these aspects. Demo quality in `Fig. 4` considers two factors described in `Sec. 5.1` (**Metrics**):
>
> > **(1) Visual fidelity**: whether the synthesized visual observations remain artifact-free and physically grounded (teleoperation and BiDemoSyn preserve raw real-world observations; DemoGen/YOTO alter the seed point cloud and introduce distortions, as shown in `Fig. 4B`).
>
> > **(2) Action/trajectory feasibility**: whether the actions can be directly executed in real hardware without failure or unstable contacts.
>
> These two dimensions correspond precisely to the input and output channels of visuomotor policies. We have added a dedicated `Appendix B.7` (**Evaluation of Demonstration Quality**) with additional clarifications.
>
> As for the concern of **ID vs. OOD settings**, **ID** means all object instances have been seen during training. The **OOD** indicates holding out a single or paired object instances entirely from the training data of per task and only presenting them during evaluation, introducing shape/height variations. This construction reflects realistic category-level generalization. In our answer to question `(A3) in Section 5.2`, we detailed the distinction and the exact protocol of OOD testing.

---

> ### Author Response · Authors · 2025-11-28
> **Response by Authors to Reviewer x4fy [Part 2]**
>
> ## W3: On MoMaGen
>
> Thank you for pointing out this relevant recent work. MoMaGen (https://arxiv.org/abs/2510.18316) was released on arXiv (Oct 2025) after the ICLR submission deadline and therefore was not included initially. We now discuss MoMaGen in the revised Related Work and provide a detailed comparison in `Appendix B.6` (**Positioning with Respect to Prior Work**). Conceptually, MoMaGen is a simulation-based long-horizon mobile+bimanual dataset generator, whereas BiDemoSyn focuses on **real-world, contact-rich, one-shot–conditioned synthesis**, which fills a different gap.
>
> ## Q1: Metrics for Demonstration Quality (Fig. 4)
>
> As detailed above (`W2`), demo quality reflects: (1) **visual fidelity** of observations, and (2) **physical feasibility** of the synthesized actions. These directly correlate with visuomotor learning performance, which is why we evaluate them qualitatively and in downstream policy success.
>
> ## Q2: Are segmentation boundaries manually annotated? Can they be automated?
>
> The segmentation requires **no explicit labels**. During kinesthetic teaching (`Appendix A.3`), human demonstrators naturally pause at meaningful task inflection points (e.g., lifting or moving after a stable grasp), creating clear boundaries between blocks. This resembles standard teleoperation and does not add overhead.
>
> We explored automatic keyframe/keypose detection, but reliably identifying semantically aligned waypoints from a single demonstration proved brittle. Hardware-assisted alternatives (such as DexCap gloves, end-effector teaching rigs like UMI/DexUMI) can automate segmentation but introduce modality-mismatch and calibration issues. We discuss these options in the revised appendix (closely after `Appendix A.3`).
>
> ## Q3: Reordering or randomizing block sequences for diversity?
>
> This is a valuable question. For contact-rich manipulation, diversity in execution style (timing, small pose variations, motion smoothness) is far less important than achieving the task reliably. Reordering blocks would artificially violate task causality (e.g., pouring before lifting the bottle). Adding synthetically diverse but semantically irrelevant variations can actually harm policy learning, as also observed in recent studies (e.g., *"Is Diversity All You Need for Scalable Robotic Manipulation?"* with the link https://arxiv.org/abs/2507.06219).
>
> Instead, we introduce diversity where it matters: object pose variations and category-level geometry variations, which meaningfully improve generalization while keeping trajectories valid. Of course, the question of what kind of data diversity the robot manipulation community needs remains open to explore, and we welcome the reviewers to continue the discussion with us. These discussions on the diversity of synthetic data were compiled and added to `Appendix B.8` (**Diversity of Synthetic Demonstrations**)
>
> ## Q4: Clarifying ID/OOD settings
>
> As addressed in `W2`, **ID** means all training object instances are present during test time. And **OOD** represents one or two object instances (with different size/shape) are completely held out from training. As shown in `Tab. 1` right, the evaluation stress-tests robustness to unseen geometry within the same task, and BiDemoSyn-trained policies degrade least under this shift.
>
> ***
>
> ## Closing Remark
>
> We thank the reviewer again for the detailed feedback. We believe the clarifications above strengthen the methodological and experimental exposition and better highlight BiDemoSyn's core contribution: a real-world, scalable, physically grounded demonstration synthesis system, addressing a critical bottleneck in data-hungry bimanual policy learning.

---

### Author Response · Authors · 2025-11-28
**Author Final Remarks**

We sincerely thank all reviewers for their constructive feedback. We substantially revised and expanded the paper to address all concerns and to further strengthen the technical clarity, empirical validation, and positioning of BiDemoSyn as a practical, scalable, real-world synthesis framework for bimanual manipulation.

## (1) System Design, Novelty, and Decomposition.
We clarified that BiDemoSyn is a real-world demonstration-synthesis paradigm rather than a composition of existing modules, and we refined the description of invariant/variant blocks and dual-arm synchronization (with new evidence showing a 22% efficiency gain and the supplementary video `new_exps_sync_motion.mp4`). We expanded `Appendix B.6` to position our decomposition strategy relative to MoMaGen and recent decomposition-based work, reinforcing the novelty and relevance of our approach.

## (2) Vision Alignment Robustness.
We clarified the difference between controlled pose-evaluation (<2% error) and real rollout orientation failures (due to mask noise), and justified our VFM + second-moment approach over heavy 6D estimators. New perturbation-robust execution videos (`new_exps_interference.mp4` and `new_exps_more_tasks.mp4`) and analysis confirm the reliability of the alignment stage across varying object poses.

## (3) Scalability: New Long-Horizon Tasks & Few-Shot Extension.
We added two new, more complex real-world tasks (*reorient+unscrew*, *unscrew+pouring*), each with 1000+ synthesized demos and successful trained policies, demonstrating scalability beyond the original six relatively simple tasks (`Appendix D.3`). A new few-shot extension further shows that additional real demos modestly improve OOD generalization, validating BiDemoSyn's extensibility (`Appendix D.4`) .

## (4) Generalization Beyond Clean Scenes.
To address concerns on simple backgrounds, we added `Appendix D.5` and `Fig. 17` showing precise VFM-based object anchoring in clutter with distractors, confirming that BiDemoSyn's perception module remains effective in more realistic and unstructured environments.

## (5) Visual Observation Generation & Policy Training.
We clarified that BiDemoSyn uses open-loop waypoint-based policies (DP3/EquiBot), where only the initial real observation is needed and intermediate states come from IK execution. We added RGB-only policy results and expanded `Appendix B.7` to better describe data-quality evaluation.

## (6) Efficiency, Failure Analysis, and Comparisons.
We clarified why real-world resets (2–3s per sample) remain efficient and higher-fidelity than point-cloud editing approaches, and added expanded failure-case analysis (`Appendix D.2`) and new comparisons with DemoGen/YOTO and simulation-based methods, addressing concerns about assumptions, robustness, and scalability.

We believe these extensive clarifications, new analyses, and new experiments significantly strengthen the paper and hope they satisfactorily address all reviewer concerns.

---

### Meta-Review · Area_Chair_Xvvu · 2026-01-06

**Summary:**

Reviewers have concerns about the system design and their novelty. The experiments lack sufficient analysis. Moreover, recent works such as MoMaGen are not compared in the original version. Moreover, the assumptions about tasks and objects are mentioned by one of the reviewer as too strong.

**Reviewer Concerns:**

Many concerns are addressed. However, the overall empirical results are still on the easy and limited side of the spectrum.
The paper is borderline for acceptance. More experiments can be considered to add to its next version.

**Reviewer Scores:**

6 4 4 8

---

### Decision · Program_Chairs · 2026-01-26

Reject